# Basal–epithelial subpopulations underlie and predict chemotherapy resistance in triple-negative breast cancer

Mohammed Inayatullah[1], Arun Mahesh [ID][1], Arran K Turnbull[2], J Michael Dixon [ID][2], Rachael Natrajan [ID][3] & Vijay K Tiwari [ID][1,4,5,6,7 ✉]

## Abstract

Triple-negative breast cancer (TNBC) is the most aggressive breast cancer subtype, characterized by extensive intratumoral heterogeneity, high metastasis, and chemoresistance, leading to poor clinical outcomes. Despite progress, the mechanistic basis of these aggressive behaviors remains poorly understood. Using single-cell and spatial transcriptome analysis, here we discovered basal epithelial subpopulations located within the stroma that exhibit chemoresistance characteristics. The subpopulations are defined by distinct signature genes that show a frequent gain in copy number and exhibit an activated epithelial-to-mesenchymal transition program. A subset of these genes can accurately predict chemotherapy response and are associated with poor prognosis. Interestingly, among these genes, elevated ITGB1 participates in enhancing intercellular signaling while ACTN1 confers a survival advantage to foster chemoresistance. Furthermore, by subjecting the transcriptional signatures to drug repurposing analysis, we find that chemoresistant tumors may benefit from distinct inhibitors in treatment-naive versus post-NAC patients. These findings shed light on the mechanistic basis of chemoresistance while providing the best-in-class biomarker to predict chemotherapy response and alternate therapeutic avenues for improved management of TNBC patients resistant to chemotherapy.

**Keywords** Genomics; Breast Cancer; EMT; Metastasis; Therapy Resistance

**Subject Categories** Cancer; Chromatin, Transcription & Genomics

## Introduction

Triple-negative breast cancer (TNBC) lacks the expression of estrogen, progesterone, and HER2 receptors, and accounts for ~20% of all breast cancer cases globally. TNBC is a highly aggressive form of breast cancer, with 40% mortality within the first 5 years of diagnosis (Bianchini et al, 2016; Charpentier and Martin, 2013; Malorni et al, 2012). Chemotherapy is currently the first line of treatment for TNBC patients. Unfortunately, nearly half of the patients develop resistance to chemotherapies, resulting in high rates of metastatic recurrence and poor survival outcomes in TNBC (Foulkes et al, 2010; Liedtke et al, 2008). Molecular signatures that can predict the response to chemotherapy are essential to inform personalized treatment and to open new avenues for targeted therapeutic intervention in cancer (Bianchini et al, 2016). For example, signature gene panels, including Oncotype DX (Bear et al, 2017; Gianni et al, 2005; Yardley et al, 2015), Endopredict (Bertucci et al, 2014) and PROSIGNA, can predict the responses of ER-positive breast cancer to chemotherapy (Prat et al, 2016). Although these tests are not effective for TNBC, increasing evidence suggests that other predictive gene signatures exist for TNBC (Fournier et al, 2019; Juul et al, 2010; Lim et al, 2020; Stover et al, 2016; Witkiewicz et al, 2014). However, none of these signatures reach a high enough prediction accuracy for clinical utility for aggressive TNBC patients.

Furthermore, previous efforts to identify chemoresistance signatures have relied on traditional bulk RNA-seq methods and have failed to capture cellular diversity profiles and signatures that are expressed specifically within the subpopulations of aggressive tumors. In recent years, single-cell sequencing technologies have emerged as powerful tools for resolving tumoral heterogeneity, reconstructing evolutionary lineages, and detecting rare subpopulations in many cancers, including TNBC (Gao et al, 2017; Kim et al, 2018; Lohr et al, 2014; Tirosh et al, 2016; Wang et al, 2014; Yuan and Sims, 2016). TNBC has been shown to have a high degree of inter- and intratumoral heterogeneity, which contributes to the disease aggressiveness, including resistance to chemotherapy (Craig et al, 2013; Houssami et al, 2011; Karaayvaz et al, 2018; Kim et al, 2018; Koren and Bentires-Alj, 2015). Unlike ER-positive cancers, eliminating the majority of cancer cells with therapy has relatively little impact on clinical outcomes in TNBC (Foulkes et al, 2010; Liedtke et al, 2008). Minor populations of metastatic and chemoresistant cells that remain after treatment are reported to

[1]Institute for Molecular Medicine, University of Southern Denmark, Odense M, Denmark. [2]Edinburgh Breast Cancer Now Research Group, Institute of Genetics and Cancer, University of Edinburgh, Western General Hospital, Edinburgh EH4 2XU, UK. [3]The Breast Cancer Now Toby Robins Research Centre, The Institute of Cancer Research, London SW3 6JB, UK. [4]Wellcome-Wolfson Institute for Experimental Medicine, School of Medicine, Dentistry & Biomedical Science, Queens University Belfast, Belfast BT9 7BL, UK. [5]Patrick G Johnston Centre for Cancer Research, Queen's University Belfast, Belfast BT9 7AE, UK. [6]Danish Institute for Advanced Study (DIAS), Odense M, Denmark. [7]Department of Clinical Genetics, Odense University Hospital, Odense C, Denmark. ✉E-mail: tiwari@health.sdu.dk

be the major contributors to disease recurrence in TNBC tumors (Kim et al, 2018). Therefore, identifying signatures that define these chemoresistant subpopulations could be a more effective strategy for targeting these cells and predicting response to Neoadjuvant chemotherapy (NAC) in TNBC.

In this study, we performed a cross-platform integrated analysis of expression profiles, derived from single-cell RNA-seq (scRNA-seq), spatial transcriptomics, bulk RNA-seq, gene expression microarray and genome sequencing data from treatment-naive and post-NAC TNBC tumors, and identified subpopulations of basal epithelial cells that spatially reside within tumor and in close vicinity to stromal compartment within the tumor microenvironment and correlate with chemoresistance. These populations exhibit activated EMT programs and are defined by a robust 101 signature gene set that is TNBC subtype-specific, upregulated in independent cohorts of TNBC residual disease, frequently associated with copy number gains and correlates with poor prognosis. These signature genes are significantly upregulated in basal and mesenchymal-like subtypes, the two most aggressive forms of TNBCs. In addition, our signature genes can accurately predict response to NAC in primary as well as advanced-stage TNBC tumors, providing a superior biomarker for stratifying TNBC patients. Notably, we found *ACTN1* to be essential for the viability of TNBC cells and an increase in ITGB1-associated intercellular signaling pathways in chemoresistant patients, which are among two of our signature genes. Further drug repurposing analysis using these gene expression signatures showed that the aggressive tumors may have sensitivity to kinase inhibitors in untreated patients while HDAC inhibitors, Disease-modifying antirheumatic drugs (DMARDs), proteasome inhibitors, and tyrosine kinase inhibitors (TKIs) will likely benefit post-NAC-treated patients with the residual disease.

# Results

## Basal epithelial subpopulations within TNBC tumors preferentially express genes associated with chemoresistance and metastasis

To investigate the intratumoral heterogeneity profile of TNBC patients, we first performed a meta-analysis of primary tumors from 25 primary TNBC patients (Fig. 1A). Here, we examined single-cell RNA-sequencing (scRNA-seq) datasets from three independent studies that had sampled primary untreated TNBC (Data ref: Chung et al, 2017; Data ref: Gulati et al, 2020; Data ref: Karaayvaz et al, 2018) (*n* = 3985 cells), and uncovered seven subpopulations of distinct cell types within the first dataset (Fig. 1B). Similarly, second and third datasets also show similar subpopulations with much higher heterogeneity in TNBC tumors compared to other breast cancer Luminal and Her2 subtypes (Fig. EV1A). We investigated whether any of these subpopulations were enriched for aggressive features such as chemoresistance and metastasis. Toward this, we utilized chemoresistance-associated gene sets, consisting of 143 genes, highly activated in residual TNBC tumors treated with neoadjuvant chemotherapy (NAC) and linked with chemotherapy resistance in cancer (Balko et al, 2012). In addition, we used an additional 49 gene signature associated with metastasis (Lawson et al, 2015). Interestingly, further analysis

showed that the basal epithelial subpopulation in primary TNBC tumors exhibited high levels of both of these signature gene sets (Figs. 1C and EV1B,C), suggesting that this specific population is associated with aggressive clinical behavior.

Next, we investigated whether these basal epithelial cells are malignant in nature, or they are non-cancerous epithelial cells. To do this, we investigated their chromosomal copy number changes (CNV) using inferCNV package (Patel et al, 2014). Here, a CNV profile of 860 TNBC cells, comprising 602 luminal epithelial cells, 188 luminal progenitor cells, and 70 basal epithelial cells was computed by comparing with 240 normal mammary epithelial cells from a previous study (Data ref: Gao et al, 2017). Our CNV analysis revealed that the majority of epithelial cells, including basal epithelial cells, exhibit a higher copy number variation profile (Fig. 1D, bottom heatmap), compared to reference normal epithelial cells (Fig. 1D, upper heatmap). In addition, the extent of CNV signal in each cell was further computed as "inferCNV score" to dissect these differences in cell-type level within TNBC epithelial populations. Plotting these scores across normal mammary and TNBC epithelial cells revealed a bimodal distribution centered around an infercnv score of 0.2, separating these groups from each other (Fig. 1E, upper left histogram). These differences were also evident at cell-type levels, where we observed higher inferCNV scores for TNBC cells including basal epithelial, compared to normal mammary epithelial cells (Fig. 1E, upper right histogram and bottom left boxplot). In addition to CNV analysis, we further validated our observations using known markers of malignancy for luminal and basal epithelial identity from cellMarker database (Hu et al, 2023) (Fig. EV1D). These observations are in line with the source study where the majority of epithelial cells were classified as malignant cells (Karaayvaz et al, 2018).

The dissociation of single cells for scRNA-seq abolishes information on the localization of specific cells within the tissue. To identify aggressive TNBC cells within the spatial context of the microenvironment, we utilized a recently published spatial transcriptomic dataset of primary tumors (Data ref: Bassiouni et al, 2023), derived from chemotherapy-treated TNBC patients with recurrence of the disease, indicative of a poor response. Here we analyzed these data and performed spatial deconvolution to annotate cell-type abundances using cell-type annotations from primary TNBC tumors (Karaayvaz et al, 2018) (data used in Fig. 1B). Interestingly, the chemoresistance-associated subpopulation of basal epithelial cells spatially located within the tumor epithelial cells and in close vicinity to the stromal compartment (Fig. 1F). These findings were consistent across spatial transcriptomes of multiple TNBC patients with recurrent disease (Fig. EV1E). The spatial arrangement of basal epithelial cells near to the stroma is not surprising as these cells are known to be juxtaposed next to the stroma and/or the basement membrane (Gusterson and Eaves, 2018).

We further identified 101 genes that defined the basal epithelial subpopulation reproducibly in at least two of the three primary TNBC scRNA-seq datasets (Fig. 1G) and considered these as "signature genes" that potentially define the specific subpopulation associated with aggressive disease (Fig. EV1B,C). The prognostic role of our signature genes was investigated using Kaplan–Meier (KM) survival analysis (Gyorffy, 2023) in independent cohorts of more than 400 TNBC patients. The 5-year relapse-free survival (RFS) was significantly reduced in the patients' with elevated

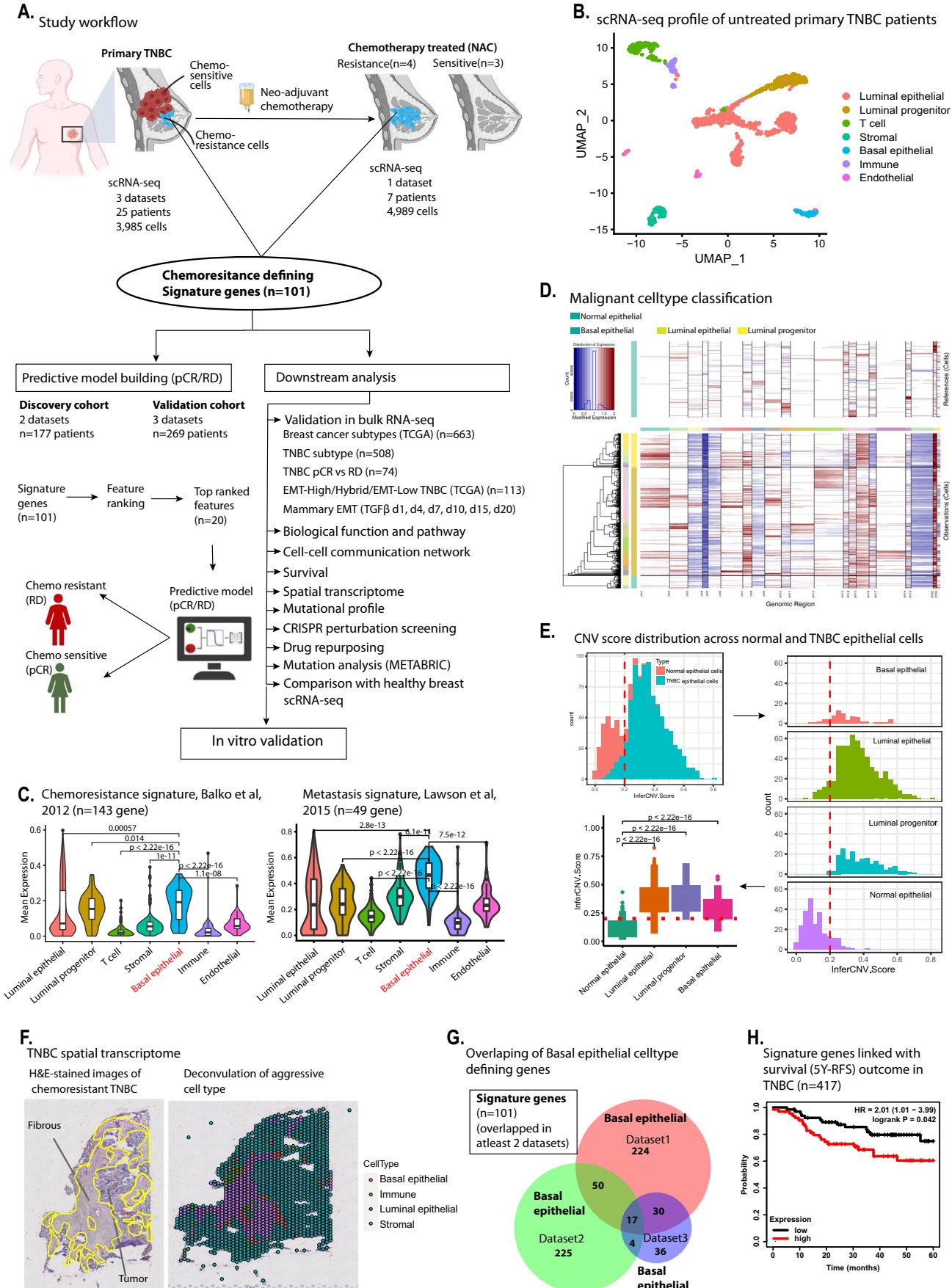

**A.** Study workflow

**B.** scRNA-seq profile of untreated primary TNBC patients

**D.** Malignant celltype classification

**E.** CNV score distribution across normal and TNBC epithelial cells

**C.** Chemoresistance signature, Balko et al, 2012 (n=143 gene)

Metastasis signature, Lawson et al, 2015 (n=49 gene)

**F.** TNBC spatial transcriptome

H&E-stained images of chemoresistant TNBC

Deconvulation of aggressive cell type

**G.** Overlaping of Basal epithelial celltype defining genes

**H.** Signature genes linked with survival (5Y-RFS) outcome in TNBC (n=417)

**Figure 1.  Single-cell transcriptomic analysis reveals cell populations associated with TNBC aggressiveness.**

(A) Schematic workflow of the study. We utilized scRNA-seq datasets of treatment-naive primary and chemotherapy-treated TNBC patients and identified small cell populations of basal epithelial cells associated with chemoresistance-like characteristics. The spatial arrangements of these aggressive cells within tissue sections of resistant TNBC tumors was identified. Further, genes defining these subpopulations were validated in bulk RNA-seq of tumors chemoresistance tumors and utilized for building a predictive model in stratifying patients with residual disease and pathological complete response. The potential drug candidates against chemoresistant cells was identified using drug repurposing approach. (B) The scRNA-seq data analysis shows cellular heterogeneity profile within six primary TNBC tumors. The genes defining each cluster were annotated against cellmarker database to assign cell-types identity to each cluster. (C) Violin plot showing expression of signature genes associated with metastasis (right plot) signatures. The expression of metastasis signature of 49 genes from Lawson et al, 2015 was plotted across each cluster. The chemoresistance signature (left plot) of 143 genes from Balko et al, 2012 expression was plotted across each cell type. In the box-and-whisker within violin plots, the horizontal lines mark the median, the box limits indicate the 25th and 75th percentiles, and the whiskers extend to 1.5× the interquartile range from the 25th and 75th percentiles. The statistical testing of expression levels of metastasis and chemoresistant genes between the cell types was performed using two-tailed unpaired Wilcoxon test in stat_compare_means() of ggpubr package. (D) The infercnv analysis of TNBC epithelial cells. The upper heatmap plot shows copy number alternation profile in healthy mammary epithelial cells. The lower heatmap plot showing CNV profile in TNBC epithelial cells. We have used total 240 healthy mammary epithelial cells to compute somatic copy number alteration in TNBC epithelial, including basal epithelial cells. Regions of chromosomal amplification manifest as blocks of red, while chromosomal deletions manifest as blue blocks, providing a visual representation of the copy number changes. (E) Copy number score, "infercnv scores" computed from inferCNV analysis of normal vs TNBC epithelial cells. The top left histogram plot shows binomial distribution of infercnv score of normal epithelial vs TNBC epithelial cells. The infercnv scores less than 0.2 defined normal epithelial cells and score greater than 0.2 defined TNBC epithelial cell types. The right histogram plot shows cell-type level infercnv score distribution. The red dotted lines shows infercnv scores threshold separating normal epithelial from TNBC epithelial cells. The lower left boxplot was used to plot to calculate statistical difference of infercnv score between the normal and TNBC epithelial cell types. In the box-and-whisker plots, the horizontal lines mark the median, the box limits indicate the 25th and 75th percentiles, and the whiskers extend to 1.5× the interquartile range from the 25th and 75th percentiles. The statistical testing of infercnv score between the normal and TNBC epithelial cells was performed using two-tailed unpaired Wilcoxon test in stat_compare_means() of ggpubr package. (F) Spatial transcriptome dataset of aggressive TNBC tumor. Left plot is an H&E-stained image of TNBC patient with recurrence within 9.4 months. The right plot shows the cell-type annotation of spot-level data. The annotation was assigned from spatial deconvolution using the primary TNBC tumor scRNA-seq dataset used in Fig. 1B. The yellow line in left plot is discriminating the fibrous cells (encircled) and tumor epithelial cells. (G) Venn diagram showing basal epithelial populations defining genes, reproducible across three datasets. Basal epithelial cell-type-defining genes overlapped between all three datasets and genes evident in at least two datasets were considered as reproducible signature genes. (H) Kaplan–Meier survival analysis plot showing correlation of signature gene expression with 5-year relapse-free survival (RFS) in TNBC patients. Analysis was performed using mean expression levels of all 101 signature genes in 417 TNBC patients using kmplotter. Source data are available online for this figure.

expression of these genes (Fig. 1H). Overall, these findings suggest that the aggressive behavior of TNBC tumors, notably therapy resistance, and metastasis, emanate from basal epithelial subpopulations, spatially reside in close vicinity to the stromal compartment within TNBC, and contribute to a poor prognosis.

## Pre-existing basal epithelial subpopulations in patients with chemoresistant TNBC

To further confirm the role of the identified signature genes in TNBC aggressiveness, we analyzed an existing scRNA-seq dataset of matched pre and post-treated TNBC tumors ($n = 4989$ cells) from three patients who responded to NAC (docetaxel and epirubicin) (chemosensitive) versus four patients who persisted with residual disease (RD) (chemoresistance) (Fig. 2A,B) (Data ref: Kim et al, 2018). The samples in this dataset were derived from core biopsies prior to NAC (0 cycle, pre-treatment), after two cycles of NAC, and from a surgical sample collected after four additional cycles of NAC in combination with bevacizumab (post treatment).

We began the analysis by quality check and correction for batch effect in these datasets (Fig. EV2A). Clustering of cells from both groups of patients revealed four and five clusters in chemosensitive and chemoresistant datasets, respectively (Fig. EV2A, bottom umap). Within the chemosensitive group, we observed completely distinct clusters of cells pre- and post chemotherapy treatment. (Fig. 2A,B, left UMAP). Notably, cells from the chemoresistant patients pre- and post treatment co-existed in the same cluster at the opposite ends on the cluster, indicative of a distinct pre-existing transcriptional programs in the resistant patients (Fig. 2A,B, right UMAP). Expression profiling of our signature genes in tumor samples from the two patient groups revealed that their expression varied between patient samples (Fig. EV2B). Importantly, however,

the chemoresistant patients as a group had a higher expression of these signature genes in untreated tumors and a further increase in expression particularly within basal epithelial cells following exposure to NAC ($P < 2.22\text{e-}16$) (Fig. 2B–D).

To investigate the role of the signature genes in clonal selection or evolution following NAC within chemoresistant patients, we obtained the clonal status of each cell using the existing data, and profiled expression levels of signature genes across the subclones. Interestingly, profiling of the signature genes ($n = 101$) in the subclonal population of cells showed higher expression in the post treatment dominant resistant clone (ClonA-Resis). Notably, expression of signature genes was also elevated in the pre-treatment clone (preTX) (Fig. 2E), indicating their possible role in defining the pre-existent clone that evolves into the dominant resistant clone upon exposure to NAC.

## Chemoresistance is associated with an active EMT program in treatment-naive tumors

To determine global transcriptional programs that are pre-existing and associated with NAC resistance in TNBC we compared gene expression profiles of pre-treated chemoresistant and chemosensitive cells using the MAST test (Finak et al, 2015). MAST identifies differentially expressed (DE) genes between two groups of cells using a tailored hurdle model which is frequently used to identify DE genes in scRNA-seq datasets (Dal Molin et al, 2017). We observed more upregulated genes ($n = 1444$ genes) than downregulated ones ($n = 566$ genes) in pre-treated chemoresistant versus chemosensitive groups (Fig. 2F). Of these DE genes, 39 genes within our signature list were upregulated and 16 were downregulated in chemoresistant groups (Fig. 2F). The upregulation of signature genes in treatment-naive groups suggests their putative

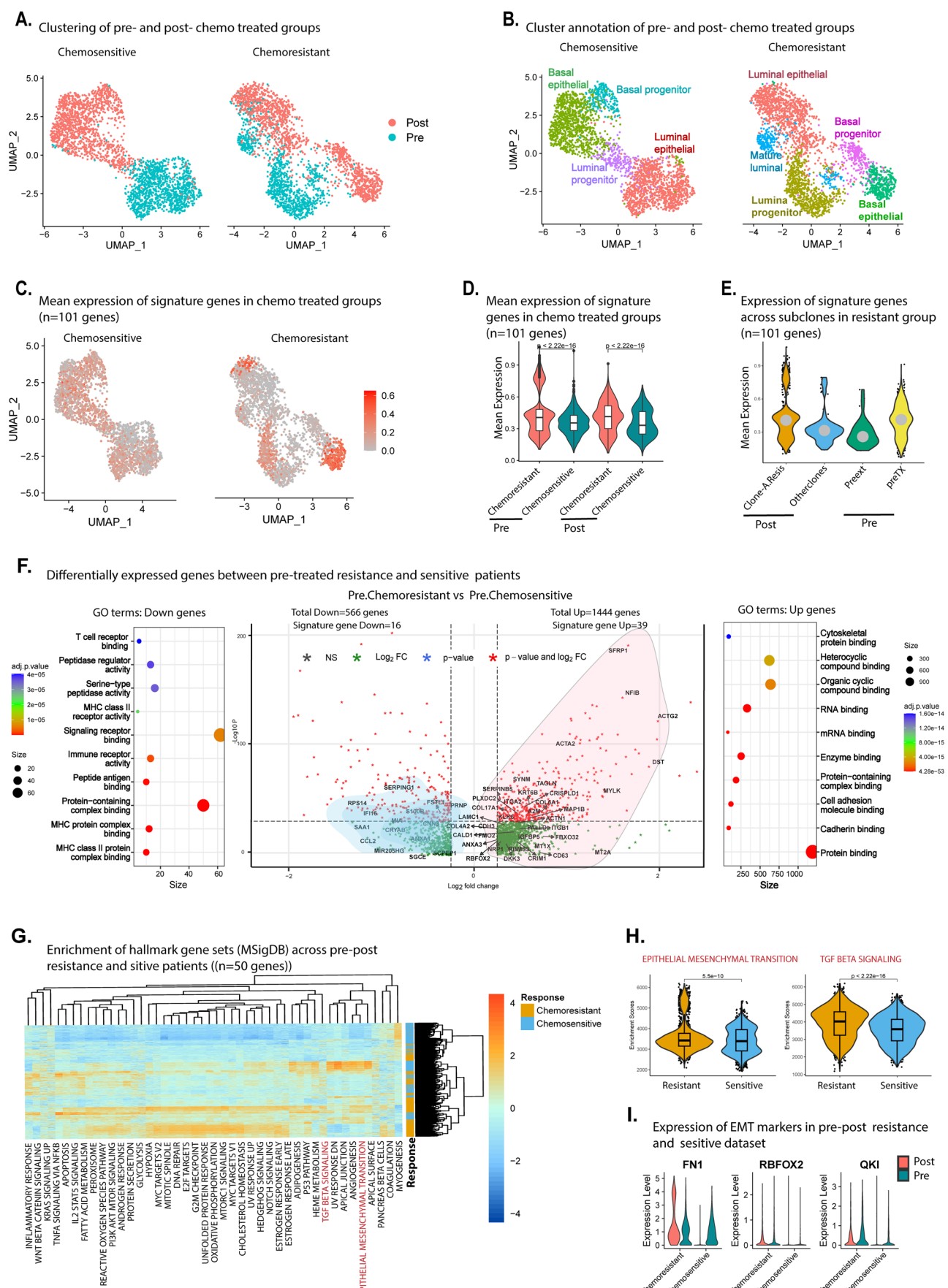

A. Clustering of pre- and post- chemo treated groups

B. Cluster annotation of pre- and post- chemo treated groups

C. Mean expression of signature genes in chemo treated groups (n=101 genes)

D. Mean expression of signature genes in chemo treated groups (n=101 genes)

E. Expression of signature genes across subclones in resistant group (n=101 genes)

F. Differentially expressed genes between pre-treated resistance and sensitive patients

Pre.Chemoresistant vs Pre.Chemosensitive

GO terms: Down genes

GO terms: Up genes

G. Enrichment of hallmark gene sets (MSigDB) across pre-post resistance and sitive patients ((n=50 genes))

H. EPITHELIAL MESENCHYMAL TRANSITION / TGF BETA SIGNALING

I. Expression of EMT markers in pre-post resistance and sesitive dataset

◄ **Figure 2. Chemoresistance is associated with an active EMT program in treatment naive tumors enriched for our signature genes.**

Expression analysis of signature genes in seven TNBC tumors (collected pre-treatment and after six cycles of NAC (docetaxel and epirubicin) (post treatment) were analyzed. (A) UMAP plot of chemosensitive and chemoresistance patients indicating pre and post-treated cells. (B) Cell-type annotation of pre and post-treated chemosensitive (n = 2497 cells) and chemoresistance (n = 2492 cells) shown in (A). (C) The UMAP plot showing mean expression profile of signature genes between the resistant and sensitive groups. (D) The mean expression profile of signature genes between the pre and post-treated resistant and sensitive groups is indicated in the violin plot. (E) Plot showing expression of signature genes across pre- and posttreatment subclones in chemoresistant patients. (D, E) In the box-and-whisker within violin plots, the horizontal lines mark the median, the box limits indicate the 25th and 75th percentiles, and the whiskers extend to 1.5× the interquartile range from the 25th and 75th percentiles. The statistical testing of expression levels of signature genes between groups was performed using two-tailed unpaired Wilcoxon test in stat_compare_means() of ggpubr package. (F) Differentially expressed genes between untreated chemosensitive and chemoresistant cells are shown in the volcano plot. Gene highlighted with names inside the volcano plot are within our 101 signature gene list. The dot plot present on each side of the volcano plot represents enrichment of gene ontology biological processes for genes that were downregulated (left panel) and upregulated (right panel) in chemoresistant versus chemosensitive cells. The significantly altered genes was identified using combined binomial and normal–theory likelihood ratio test in MAST. (G) The Heatmap shows enrichment of 50 hallmark signature pathways in chemosensitive and chemoresistance subpopulations using the Molecular Signatures Database (MSigDB). The pathways highlighted in the red and orange boxes are enriched in chemoresistance and chemosensitive subpopulations. (H) The violin plot shows enrichment of EMT and TGF-b pathway genes in resistant and sensitive group of cells. In the box-and-whisker within violin plots, the horizontal lines mark the median, the box limits indicate the 25th and 75th percentiles, and the whiskers extend to 1.5× the interquartile range from the 25th and 75th percentiles. The significance test was performed using two-tailed unpaired Wilcoxon test. (I) Violin plot showing expression of hallmark EMT genes across pre and post-treated chemoresistance and chemosensitive patients. Source data are available online for this figure.

role in attaining intrinsic chemoresistance characteristics in TNBC. Further biological function analysis revealed that, the upregulated genes in pre-treatment chemoresistant cells were enriched for GO terms associated with RNA binding, protein binding and cell adhesion, and cytoskeleton binding, which are key processes in cell-fate changes such as Epithelial-to-Mesenchymal Transition (Wheelock et al, 2008; Yilmaz and Christofori, 2009) (Fig. 2F, right dot plot). On the other hand, downregulated genes in untreated chemoresistant cells were enriched for immune response-related processes such as MHC Class II activity, Immune receptor activity, and T-cell receptor binding (Fig. 2F, left dot plot). These results are in line with earlier reports that an increase in immune response activity correlates with a better response of TNBC patients to NAC (Denkert et al, 2010; Garcia-Teijido et al, 2016; Ladoire et al, 2011).

To further delineate functional pathways operating in these subpopulations associated with aggressive clinical behaviors, we investigated pathway enrichment across pre- and posttreatment chemosensitive and chemoresistant cells based on the 50 hallmark gene sets available from The Molecular Signatures Database (MSigDB) hallmark gene set collection (MsigDB) (Liberzon et al, 2015). This analysis revealed that distinct categories of pathways are enriched in chemoresistance versus chemosensitive subpopulations (Fig. 2G). The epithelial-to-mesenchymal transition (EMT) is known to confer chemoresistance and is linked to aggressive behavior in many cancers including TNBC (Hong et al, 2018; Jang et al, 2015; Luo et al, 2015; Xu et al, 2018). Consequently, most breast cancer deaths (90%) are caused by tumor invasion and metastasis, which are the two key features related to the EMT (Felipe Lima et al, 2016). Interestingly, pathways associated with EMT, including TGFβ signaling, were among the pathways activated in chemoresistant subpopulations (Fig. 2H). In line with these observations, the established EMT markers FN1, RBFOX2, and QKI had elevated expression in chemoresistant versus chemosensitive patient samples, both pre- and post treatment (Fig. 2I). These observations are consistent with a correlation between the upregulation of EMT related genes and resistance to chemotherapy (Kim et al, 2018). We were unable to detect a few other hallmark EMT genes, such as the transcription factors *ZEB1/2*, *TWIST1/2*, and *SNAI1/2*, potentially due to known limitations of scRNA-seq (Lambert et al, 2018; Pokhilko et al, 2021).

## The basal epithelial signature genes are elevated during EMT and in EMT-high tumors

To further characterize the 101 signature genes that define the basal epithelial subpopulation, we performed a GO analysis, which revealed enrichment for EMT-like processes such as wound healing, extracellular matrix organization and cell migration (Fig. 3A). Recent data have highlighted that the different molecular subtypes of TNBC are associated with differential response rates to specific therapies such as the anti-PD-L1 immunotherapy Atezolizumab (Emens et al, 2021; Garrido-Castro et al, 2019). We therefore investigated the correlation of our signature genes within each molecular subtype of TNBC. For this, we obtained FPKM normalized expression levels of these signature genes from 508 TNBCs transcriptomes (METABRIC = 325 (Data ref: Curtis et al, 2012); TCGA-BRCA = 183 (Data ref: Cancer Genome Atlas Research N et al, 2013) and plotted them across four molecular subtypes i.e luminal androgen receptor (LAR), Basal-like 1 (BL1), basal-like 2 (BL2) and mesenchymal-like (M) of TNBCs. Interestingly, our signature genes were significantly elevated in basal-like and mesenchymal-like subtypes compared to LAR subtypes (Fig. 3B), consistent with the activated EMT programs and worse survival outcomes of basal-like and mesenchymal-like subtypes (Liu et al, 2016; Park et al, 2020; Yin et al, 2020). Our further analysis of the large TCGA breast cancer subtype cohort showed that these genes are expressed at significantly higher levels in TNBC tumors compared to luminal (ER-positive) and HER2-enriched breast cancers and hence suggests a key role in disease pathogenesis in TNBC subtype (Fig. 3C).

Given the limitations of the scRNA-seq technology in detecting genes with low expression levels, we analyzed a recently published bulk RNA-seq of 74 samples from TNBC patients who were treated with NAC (AC Adriamycin (Doxorubicin) + Cyclophosphamide, T Taxol (Docetaxel), H Herceptin (Trastuzumab)) and had a known treatment outcome, namely pathological complete response (pCR) and residual disease (RD) status (Data ref: Park et al, 2020). In support of our observations, our signature genes showed significantly higher expression levels in patients with residual disease (Fig. 3D).

We next investigated whether our signature genes are uniformly activated across TNBC or reflect aggressive disease with mesenchymal characteristics. To do this, we classified TCGA TNBC samples

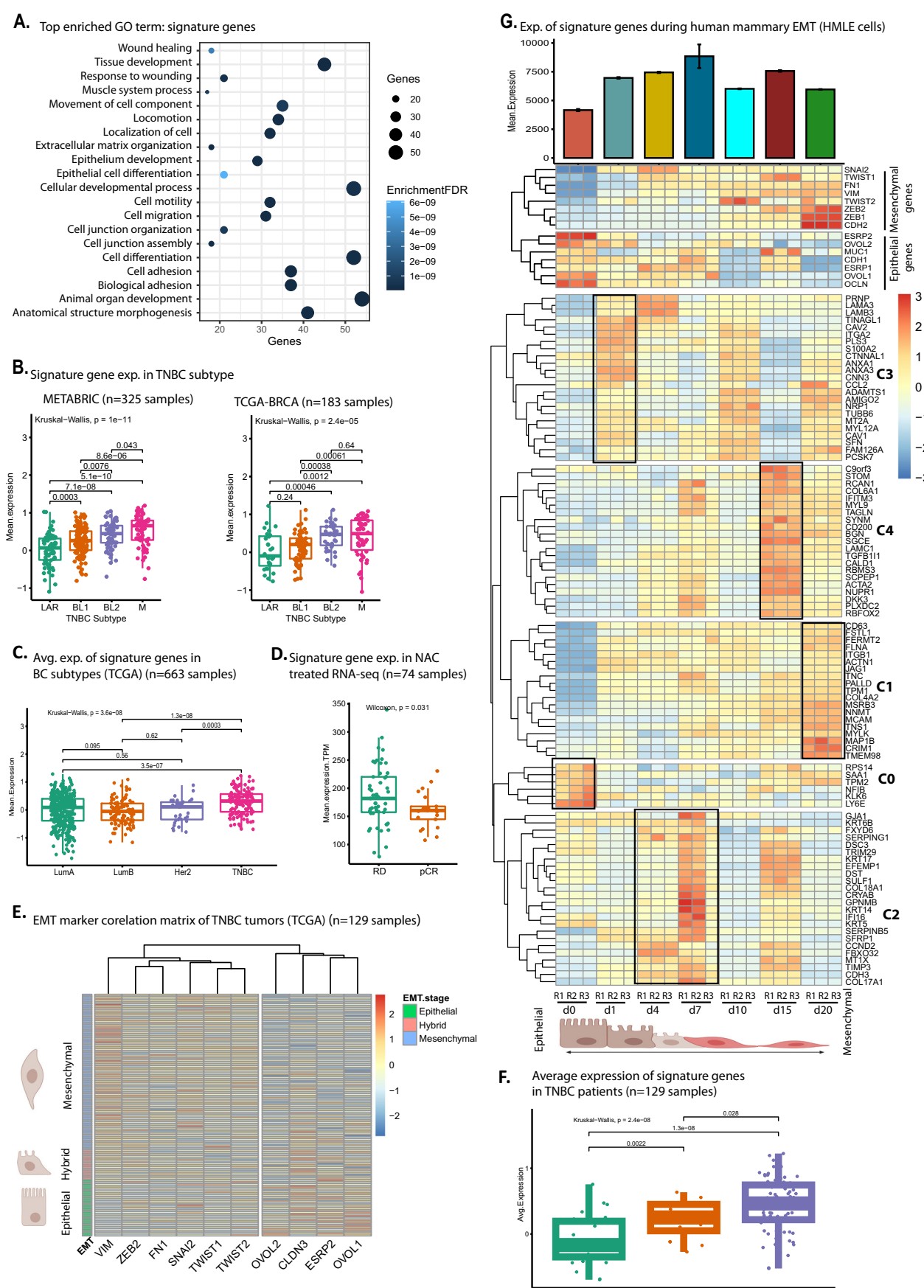

**A.** Top enriched GO term: signature genes

**B.** Signature gene exp. in TNBC subtype

**C.** Avg. exp. of signature genes in BC subtypes (TCGA) (n=663 samples)

**D.** Signature gene exp. in NAC treated RNA-seq (n=74 samples)

**E.** EMT marker corelation matrix of TNBC tumors (TCGA) (n=129 samples)

**F.** Average expression of signature genes in TNBC patients (n=129 samples)

**G.** Exp. of signature genes during human mammary EMT (HMLE cells)

◄ **Figure 3. Chemoresistance-associated signature genes are induced during EMT and are expressed at higher levels in EMT-high TNBC tumors.**

Chemoresistance-associated signature genes are TNBC subtype-specific and have activated EMT programs. (A) The dot plot shows gene ontology analysis of our signature genes enriched for biological processes associated with EMT. (B) The boxplot showing the mean expression of our signature genes across molecular subtypes of TNBC patients. The mean expression levels of our signature genes were plotted across TNBC molecular subtype luminal androgen receptor (LAR), basal-like 1 (BL1), basal-like 2 (BL2), and mesenchymal (M) in two independent cohorts i.e., METABARIC and TCGA-BRCA cohorts. (C) Boxplot of TCGA breast cancer cohort showing mean expression of signature genes across different subtypes of breast cancer. Expression profile of signature genes extracted from TCGA breast cancer cohort and each tumor classified into four subtypes i.e., LuminalA, LuminalB, HER2, and TNBC based on ER, PR, HER2 status. Next, the average expression of our signature genes was plotted across subtypes of breast cancers. (D) Boxplot showing mean expression of signature genes in bulk RNA-seq datasets of 74 chemotherapy-treated TNBC patients. (E) Expression of signature genes in EMT-High, Hybrid and EMT-Low TNBC tumors. Heatmap showing the classification of TCGA TNBC tumors into EMT-high (Mesenchymal), Hybrid and EMT-low (Epithelial) groups based on the expression of hallmark epithelial (4 genes) and mesenchymal (6 genes) genes. (F) Boxplot showing average expression of signature genes in EMT-High, Hybrid and EMT-Low TNBC (TCGA) cohort. (B–D, F) In the box-and-whisker plots, the horizontal lines mark the median, the box limits indicate the 25th and 75th percentiles, and the whiskers extend to 1.5× the interquartile range from the 25th and 75th percentiles. The statistical testing of expression levels of signature genes between the groups was performed using two-tailed unpaired Wilcoxon test in stat_compare_means() of ggpubr package. (G) The Heatmap showing expression dynamics of signature genes across different EMT timepoints (TGFβ treated) of mammary epithelial cells (HMLE) RNA-seq. The time point labeled with d0 are untreated and d1-d20 are different EMT timepoints treated with TGF-b from day 1 to day 20. The upper cluster represents the expression of hallmark EMT genes during TGFβ-induced EMT. The lower heatmap shows a cluster of genes represented by C is on the right side of the heatmap representing EMT induction time-specific genes. The mean expression of signature genes during TGFβ induced EMT in HMLE cells is shown on the top of the barplot. The Error bar plotted based on the standard deviation calculated using three replicates of each timepoints. (A–F) The "n" represents total number of samples taken for the analysis. Source data are available online for this figure.

($n = 129$) (Data ref: Cancer Genome Atlas Research N et al, 2013) into EMT-Low (epithelial), hybrid-EMT (Hybrid) and EMT-High (mesenchymal) groups based on the expression of 10 established EMT markers (Fig. 3E). Interestingly, our signature genes had significantly elevated expression in mesenchymal samples compared to epithelial samples (Fig. 3F).

To examine the involvement of these genes in EMT progression, we experimentally induced EMT in vitro by treating immortalized human mammary epithelial cells (HMLEs) with TGFβ and performed RNA-whole transcriptome sequencing at different timepoints representing early, mid and late EMT. We observed altered expression of hallmark EMT genes over time, verifying our approach (Fig. 3G, shown in the upper cluster of the heatmap). Interestingly, we observed increased expression of almost all our identified signature genes (96 out of 101) during EMT, confirming that these genes are truly associated with mammary EMT (Fig. 3G). Distinct subsets of these genes (highlighted with C0-C4 clusters) were induced at different timepoints during EMT, which might underlie their roles during EMT progression.

## Signature genes are frequently amplified in basal-like TNBC tumors

Chemoresistance in TNBC is reported to be the result of clonal selection and clonal evolution via changes in genomic landscape, such as acquired mutations and copy number variations (Almendro et al, 2014; Balko et al, 2014; Balko et al, 2016; Navin, 2014). Therefore, to unravel the genomic landscape of our signature genes, we performed mutational and copy number variation analysis on a TNBC cohort. We obtained the mutational and copy number alteration profiles of TNBC patients from METABRIC ($n = 209$) (Data ref: Curtis et al, 2012) and analyzed our signature genes in chemotherapy-treated and untreated patients. Noticeably, the analysis showed a gain in copy number of the majority of our signature genes across all patients, and a particularly significant gain in chemotherapy-treated patients compared to the untreated ones. Among these, *FBXO32* (frequency: 28–40%), *LY6E* (frequency: 12–36%), and *IFI16* (frequency: 7–27%) genes were the most frequently altered and showed gain in copy number in patients treated with chemotherapy (Fig. 4A). Interestingly, we

previously showed that FBXO32 is essential for conferring the microenvironment that drives tumor aggressiveness (Sahu et al, 2017). These results suggest that changes in the expression of these signature genes could be governed at the genomic level, and an increased copy number in chemotherapy-treated TNBC patients could further increase expression levels.

Genetic changes, including copy number alterations (CNAs), can not only drive tumor progression but also can discern tumor subtypes with distinct characteristics (Jiang et al, 2019). We next profiled the expression of our signature genes among six previously defined CNA subtypes of TNBC (Data ref: Jiang et al, 2019) and found that it is elevated in four CNA subtypes (Fig. 4B, left boxplot). These CNA subtypes represent CNA subtype 1, frequent 9p23 amplification (*Chr9p23 amp*); CNA subtype 2, frequent 12p13 amplification (*Chr12p13 amp*); CNA subtype 3, frequent Chr13q34 amplifications (*Chr13q34 amp*); CNA subtype 4, frequent Chr20q13 amplification (*Chr20q13 amp*); CNA subtype 5, frequent Chr8p21 loss (*Chr8p21 del*); and CNA subtype 6, somatic CNA lacking a CN cluster but with low chromosomal instability (CIN) (*low CIN*). Interestingly, these CNA subtypes are more frequently amplified in tumors of basal-like subtype (~85% of tumors), than LAR and MES TNBC (Fig. 4B, right pie chart). We further investigated the expression levels of our signature genes in four TNBC mutation type categories defined by Jiang et al (Jiang et al, 2019): mutation subtype 1, which is defined by enrichment of APOBEC-related genes (APOBEC); mutation subtype 2, which was highlighted by homologous recombination deficiency (HRD)-related genes (HRD); mutation subtype 3, with clock-like signatures genes (clock-like); and mutation subtype 4, with no dominant signature (mixed). We observed elevated expression in the homologous recombination deficiency (HRD) subtype (Fig. 4C, left boxplot), which also is mostly the basal-like subtype compared to LAR and MES subtypes (Fig. 4C, right pie chart). The HRD type is a copy number-based biomarker facilitating the identification of patients who might respond to DNA-damaging agents (Telli et al, 2016). Our signature genes were elevated in patients with HRD CNA subtype, suggesting its association with patients with a decreased ability to repair double-strand DNA breaks. HRD has been reported to be associated with response to standard NAC in TNBC patients (Telli et al, 2018). Overall, these findings indicate

**A.**

Mutation profile of signature genes in TNBC cohorts (METABRIC, n=209)

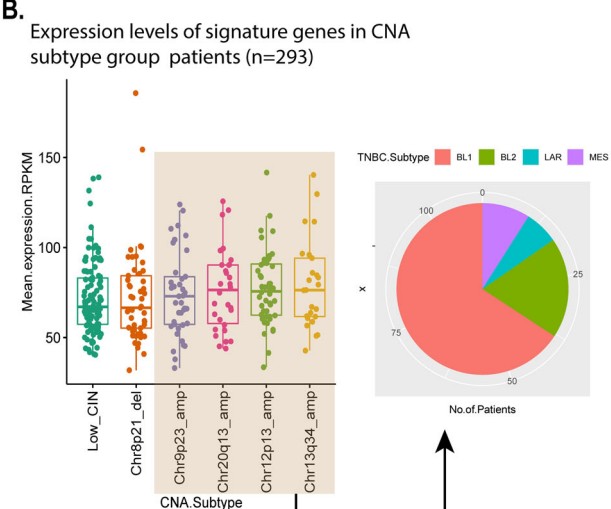

**B.**

Expression levels of signature genes in CNA subtype group patients (n=293)

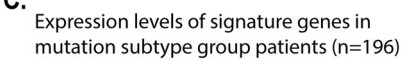

**C.**

Expression levels of signature genes in mutation subtype group patients (n=196)

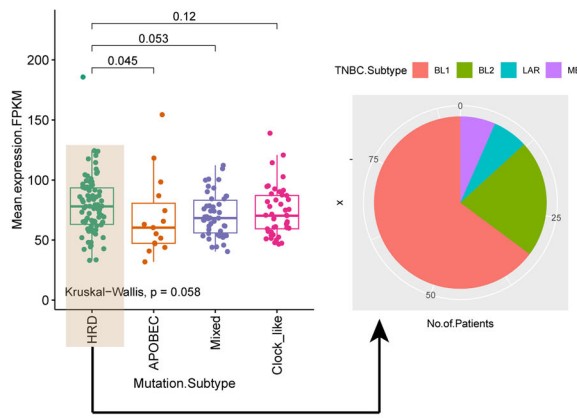

◄ **Figure 4. Elevated expression of signature genes in TNBC associated with an increased copy numbers, particularly in HRD subtype with basal-like characteristics.**

Signature genes elevated in tumors of HRD subtype with increase copy numbers changes. (A) Oncoprint heatmap showing mutations and copy number alteration in METABARIC cohort. The left and right heatmap shows alternations in signature genes through deep deletion, amplification, missense, and truncating mutations in both chemotherapy-treated and -untreated TNBC patients. The bar graph in the center represents the frequency of mutations of each gene in TNBC patients. The boxplot in the center shows overall mutational frequency between chemotherapy-treated and untreated tumors. The significance test between chemotherapy-treated and untreated groups was performed using one-tailed paired $t$ test. (B) The boxplot showing the expression of our signature genes among six previously defined CNA subtypes of TNBC (Jiang et al, 2019). (C) Boxplot showing mean expression of signature genes in four mutational subtypes of TNBC patients, homologous recombination deficiency (HRD), APOBEC, clock-like and mixed. HRD mutation type is defined by alteration in HRD-related genes; APOBEC type is defined by a mutation in APOBEC-related genes; clock-like is defined by a genetic alteration in clock-like genes, and mixed is defined by no dominant gene signature. The significance test between mutation subtypes was performed using two-tailed unpaired Wilcoxon test. (A, C) In the box-and-whisker plots, the horizontal lines mark the median, the box limits indicate the 25th and 75th percentiles, and the whiskers extend to 1.5× the interquartile range from the 25th and 75th percentiles. Source data are available online for this figure.

that our signature genes are not only deregulated at the transcriptional level but also frequently show a gain in copy number, particularly in patients with basal-like subtypes with dysfunctional DNA repair mechanisms.

## A best-in-class multi-gene classifier to predict chemotherapy response in TNBC

Pathological complete response (pCR) is the key surrogate marker for long-term prognoses, such as disease-free survival and overall survival, for TNBC patients (Hahnen et al, 2017; Liedtke et al, 2008; von Minckwitz et al, 2012). However, the field lacks clinically useful predictors of pCR for TNBC (Lehmann et al, 2016; Louie and Sevigny, 2017; Mark et al, 2017; Nakashoji et al, 2017; Santuario-Facio et al, 2017). Although molecular tests can guide treatment for estrogen receptor (ER) or HER2+ tumors, no such tests exist to stratify TNBC and inform therapeutic strategies (Bear et al, 2017; Bertucci et al, 2014; Prat et al, 2016; Yardley et al, 2015). To address this unmet clinical need, we developed a predictive model that can stratify pCR or residual disease (RD) to standard NAC in TNBC (Fig. 5A). Briefly, we performed a meta-analysis of five independent breast cancer studies (Data ref: Hatzis et al, 2011), (Data ref: Hatzis et al, 2011), (Data ref: Shi et al, 2010), (Data ref: Horak et al, 2013) and (Data ref: Tabchy et al, 2010), and examining the expression of our 101 signature genes in samples from 446 TNBC patients who were treated with standard NAC (e.g., taxane, anthracycline, cyclophosphamide, 5-fluorouracil), and whose treatment outcomes, pCR and RD were known (Fig. 5B).

To build a predictive model, we subjected our 101 signature genes to Lasso and Elastic-Net Regularized Generalized Linear Models using the glmnet package. The powerful built-in feature selection capability of glmnet allowed us to narrow down the list to 20 genes with the highest predictive power from our signature set (Fig. 5C). Before subjecting these genes to model building, we profiled their expression on spatial transcriptome dataset and found their high enrichment in basal cells in recurrent TNBC patients (Fig. 5D), which we showed earlier to be linked with chemoresistance (Fig. 1F). For model building, we used the tenfold cross-validation statistical method to estimate the skill of the model on new data. In the $k$-fold cross-validation method, a model performs the fitting procedure a total of ten times, with each fit performed on a training set comprising a random 90% of the total training set and the remaining 10% for validation. Subsequently, we used these 20 genes to build a predictive model that we fit to a generalized linear model (Fig. EV2D,E).

We used receiver operating characteristic (ROC) curve estimates to visualize the performance of a classification model

at all classification thresholds. The probability of classifying a true positive, inferred from the ROC area under the curve (AUC), was 90.3%, indicating that our model based on 20 genes shows excellent discrimination ability between the pCR and RD patients (Fig. 5E). Furthermore, our model showed high performance on three independent validation cohorts (GSE20194-GSE20271 AUC = 88.8% and AUC = 86.0% for GSE41998) (Fig. 5F) demonstrating the robustness of the signature genes. The discriminative power was significantly decreased if a single gene (AUC = 85.8%) or a set of genes (AUC = 77.2%) were removed from the model (Fig. 5G), indicating a strong combinatorial accuracy. Furthermore, our gene panel predicted chemotherapy response substantially better than five existing gene signatures (Fournier et al, 2019; Juul et al, 2010; Lim et al, 2020; Stover et al, 2016; Witkiewicz et al, 2014) (Fig. 5H). Therefore, our multi-gene signature is currently the "best-in-class" given its higher discriminative ability in a large cohort of pCR and RD patients; and superior performance over published signatures in predicting the chemotherapy response. The Kaplan–Meier survival analysis showed that higher expression of these 20 genes was associated with significantly reduced 5-year survival in TNBC patients (HR = 1.71 (1.24–2.35), $P = 0.00093$) (Fig. 5I).

## Chemoresistant TNBC cells exhibit enhanced intercellular communication

Our spatial analysis showed the existence of aggressive cells within the tumor and in close vicinity to non-tumor cells, which are known to interact with each other (Kaminska et al, 2015; Ungefroren et al, 2011). We therefore hypothesized that the aggressive basal cells engage in communication with other cell types within the TME to confer chemoresistance in TNBC. To examine cell–cell communication, we applied CellChat which quantitatively characterizes and compares the inferred cell–cell communication network, based on the average expression of the ligands and receptors in cell populations. We first examined the scRNA-seq of treatment-naive TNBC tumors (Data ref: Karaayvaz et al, 2018) (Fig. 6A, the dataset used in Fig. 1B) and found that cells of basal epithelial type exhibit strong intercellular communication with stromal cells (Fig. 6B), which fits with our observations from spatial transcriptomes that they spatially reside in close vicinity to the stromal cells (Fig. 6C). In particular, Insulin-like growth factor (IGF) signaling and LAMININ signaling pathways supported intercellular communication in these TNBC tumors (Fig. 6D). Interestingly, these pathways employ a top-ranked signature gene from the 20-panel list, ITGB1 (Fig. 6E).

**A.**

Signature gene based predictive model building workflow

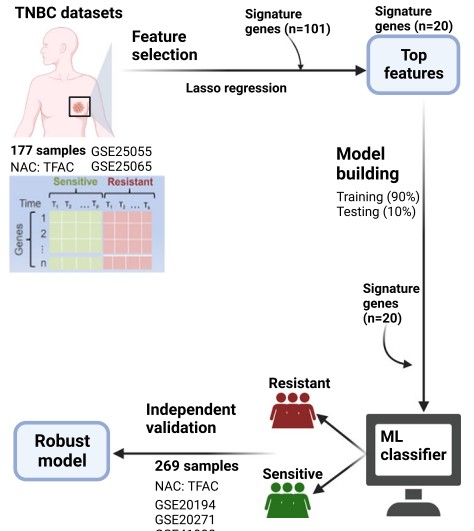

**B.**

Clinical details of samples used for model building and validation

| | Classifier building | | | External validation | | | Total |
|---|---|---|---|---|---|---|---|
| Dataset | GSE25055 | GSE25065 | Total | GSE20194 | GSE20271 | GSE41998 | Total |
| No. of patients | 121 | 56 | 177 | 71 | 59 | 139 | 269 | 446 |
| Response | | | | | | | | |
| pCR | 40 | 19 | 59 | 25 | 13 | 109 | 147 | 206 |
| RD | 81 | 37 | 118 | 46 | 46 | 30 | 122 | 240 |
| NAC | TFAC/ACT | AT | | TFAC | FAC/TFAC | TAC/I | | |
| Age. Years | | | | | | | | |
| <=50 | 68 | 33 | 101 | 38 | 28 | 77 | 143 | 244 |
| >50 | 53 | 23 | 76 | 33 | 31 | 62 | 126 | 202 |
| Tumore stage | | | | | | | | |
| T0-1 | 7 | 1 | 8 | 6 | 1 | 1 | 8 | 16 |
| T2 | 58 | 24 | 82 | 33 | 23 | 83 | 139 | 221 |
| T3 | 33 | 26 | 59 | 14 | 15 | 55 | 84 | 143 |
| T4 | 23 | 5 | 28 | 18 | 20 | 0 | 38 | 66 |
| Nodal status | | | | | | | | |
| N0 | 24 | 19 | 43 | 11 | 18 | NA | 29 | 72 |
| N1 | 61 | 24 | 85 | 36 | 24 | NA | 60 | 145 |
| N2 | 20 | 9 | 29 | 13 | 14 | NA | 27 | 56 |
| N3 | 16 | 4 | 20 | 11 | 3 | NA | 14 | 34 |
| Grade | | | | | | | | |
| G1 | | 2 | 2 | | 1 | NA | 1 | 3 |
| G2 | 16 | 5 | 21 | 11 | 8 | NA | 19 | 40 |
| G3 | 93 | 43 | 136 | 53 | 35 | NA | 88 | 224 |
| G4 | 9 | | 9 | | | NA | 0 | 9 |
| Unknown | 3 | 6 | 9 | 7 | 15 | NA | 22 | 31 |
| Reference (PMID) | 21558518 | 21558518 | | 20676074 | 20829329 | 23340299 | | |

pCR: Pathological complete response; RD: residual disease;
NAC: Neoadjuvant chemotherapy; T: Taxane; F: fluorouracil;
A:Anthracycline; C: Cyclophosphamide; I: Ixabepilone

**D.**

20 gene expression on spatial transcriptome dataset

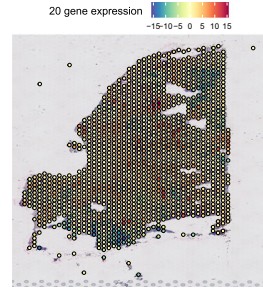

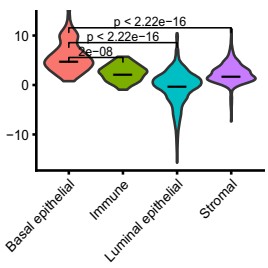

**C.**

Gene ranking for top feature identification

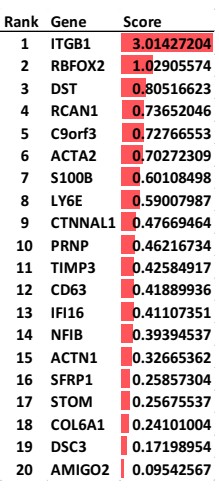

| Rank | Gene | Score |
|---|---|---|
| 1 | ITGB1 | 3.01427204 |
| 2 | RBFOX2 | 1.02905574 |
| 3 | DST | 0.80516623 |
| 4 | RCAN1 | 0.73652046 |
| 5 | C9orf3 | 0.72766553 |
| 6 | ACTA2 | 0.70272309 |
| 7 | S100B | 0.60108498 |
| 8 | LY6E | 0.59007987 |
| 9 | CTNNAL1 | 0.47669464 |
| 10 | PRNP | 0.46216734 |
| 11 | TIMP3 | 0.42584917 |
| 12 | CD63 | 0.41889936 |
| 13 | IFI16 | 0.41107351 |
| 14 | NFIB | 0.39394537 |
| 15 | ACTN1 | 0.32665362 |
| 16 | SFRP1 | 0.25857304 |
| 17 | STOM | 0.25675537 |
| 18 | COL6A1 | 0.24101004 |
| 19 | DSC3 | 0.17198954 |
| 20 | AMIGO2 | 0.09542567 |

**E.**

Discovery cohort (n= 177)

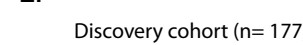

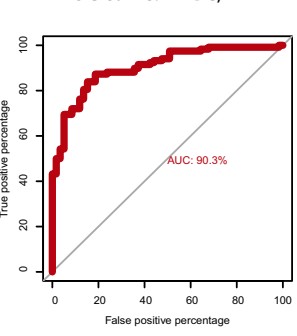

ROC curve: TNBC, n=177

AUC: 90.3%

**F.**

Validation cohort (independent validation, n= 269)

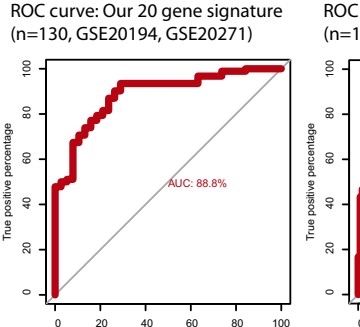

ROC curve: Our 20 gene signature (n=130, GSE20194, GSE20271)

AUC: 88.8%

ROC curve: Our 20 gene signature (n=139, NCT00455533 trial)

AUC: 86.0%

**G.**

ROC curve: QUB combinatory power

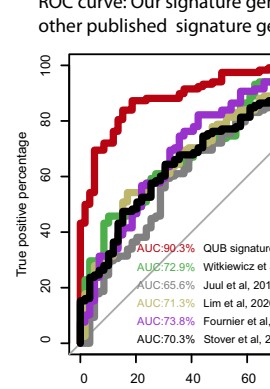

AUC: 90.3%
AUC: 85.8%
AUC: 77.2%

All 20 gene
19 genes
15 genes

**H.**

Model performance comparison with published signatures

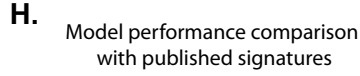

ROC curve: Our signature genes and other published signature genes

AUC:90.3% QUB signature gene (20 genes)
AUC:72.9% Witkiewicz et al, 2014 (9 genes)
AUC:65.6% Juul et al, 2010 (6 genes)
AUC:71.3% Lim et al, 2020 (15 genes)
AUC:73.8% Fournier et al, 2019 (16 genes)
AUC:70.3% Stover et al, 2016 (18 genes)

**I.**

Relapse free survival (TNBC patients)

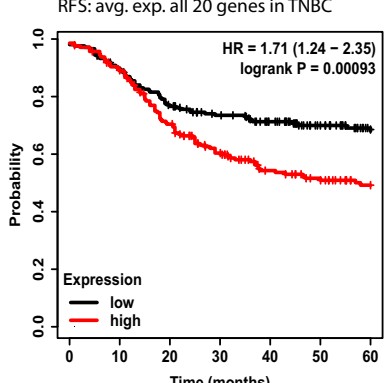

RFS: avg. exp. all 20 genes in TNBC

HR = 1.71 (1.24 – 2.35)
logrank P = 0.00093

Expression
low
high

◀ **Figure 5. A 20-gene panel can accurately predict chemotherapy response in TNBC patients.**

The predictive model development and validation with 20 gene expression levels using the least absolute shrinkage and selection operator (LASSO) regression method. (A) Predictive model building workflow. (B) Clinical details of TNBC samples used for model building and validation. (C) Ranking of genes based on the lambda score obtained from glmnet. (D) Mean expression of 20 genes on the spatial transcriptome dataset. The bottom violin plot shows mean expression of 20 genes within cell types of spatial transcriptome dataset. (E) The receiver operating characteristic (ROC) curves of the discovery cohort TNBC dataset ($n = 177$, 2 datasets—GSE25055, GSE25065) for pCR and RD prediction using 20 gene expressions as a feature. (F) The ROC curves of the TNBC validation dataset ($n = 130$, 2 datasets—GSE20194, GSE20271) and NCT00455533 trial dataset (GSE41998, $n = 139$) for pCR and RD prediction using 20 gene expressions as a feature. (G) The ROC curves were generated for all 20 gene features (red); after the removal of one gene (cadetblue); and the removal of 5 genes (purple) to assess the impact of the combination of genes on the model performance. The ROC curve highlighted with the red line used all 20 genes, whereas the model with reduced gene sets (highlighted with cadetblue and purple line) showed a difference in discriminating ability in predicting pCR vs RD in TNBC. (H) The ROC curve shows the comparative performance of our model (QUB signature gene panel) with five published signatures. Here we used the same cohort for all five gene signatures to compare the performance of our model with the published ones. (I) Kaplan–Meier survival analysis plot showing correlation of 20 gene expression with 5-year relapse-free survival (RFS) in TNBC (top) ($n = 417$). Source data are available online for this figure.

Next, we investigated whether intercellular interactions observed in primary tumors show any differences between resistant and sensitive cells upon exposure to chemotherapy (Fig. 6F,G). Therefore, we performed cellchat analysis of post-NAC-treated chemoresistant and chemosensitive cells. Interestingly, we observed a substantially higher (over twofold more) number of cell–cell communications in chemoresistant cells compared to chemosensitive cells (Fig. 6H), particularly driven by basal epithelial cells (Fig. 6I). Notably, distinct signaling pathways were enriched in chemoresistant versus chemosensitive TNBC tumors (Fig. 6J). Interestingly, similar to primary tumors, post-treated TNBC cells show stronger cell–cell communication of LAMININ signaling pathways (Fig. 6K), which utilize one of our signature genes ITGB1 (Fig. EV3A). These results also imply that a high expression of our signature gene could result in increased intercellular signaling through Tyrosin-kinase activity (i.e., IGF signaling) and LAMININ signaling cascades and thereby contribute to aggressive phenotypes and chemoresistance in TNBC tumors.

## Drug repurposing analysis identifies FDA-approved drugs to overcome TNBC chemoresistance

Given the high attrition rates, substantial costs, and slow pace of new drug discovery and development, repurposing of 'old' drugs to treat both common and rare diseases including cancers is increasingly becoming attractive (Nosengo, 2016). It offers several advantages, for example, it involves the use of de-risked compounds, with potentially lower overall development costs, lower risk of failure, and shorter development timelines (Ashburn and Thor, 2004). A key step in drug repurposing is to identify a potential candidate molecule using molecular docking, genome-wide association studies, and pathway or network mapping. However, these approaches offer limited potential as they are built on traditional molecular methods such as total RNA-seq and fail to account for tissue heterogeneity for diverse cell populations within the diseased tissue. Here, we attempted to overcome this problem in the context of TNBC chemoresistance using a recently published computational algorithm ASGARD (He et al, 2023) that ranks recommend FDA-approved drugs against cell populations using scRNA-seq datasets. ASGARD uses the differentially expressed genes (in our case, between resistant and sensitive) as inputs to identify drugs that can significantly (single-cluster FDR < 0.05) reverse their expression levels using the L1000 drug response dataset as a reference.

Our drug repurposing analysis using ASGARD against pre- and post-NAC-treated chemoresistant cell populations identified different sets of FDA-approved drugs that may have sensitivity towards chemoresistant cell populations in untreated and treated TNBC tumors (Fig. 7A,B). For example, pre-treated resistant cells showed tyrosine kinase inhibitors (TKI) family as top drug candidates i.e., Pazopanib, Fostamatinib, Crizotinib, etc. (Fig. 7B, top dot plot), while the top candidates for posttreatment resistant cells showed another set of top-ranked drugs as candidates such as HDAC inhibitor (Vorinostat), DMARDs (Auranofin), proteasome inhibitor (Ixazomib) and TKIs (Neratinib, Crizotinib, Fostamatinib, and Pazopanib) drug types (Fig. 7B, bottom dot plot). This analysis further identified drugs that may be more effective in a combination (Fig. 7B, right dot plot). The TKIs as a potential alternative drug for aggressive TNBC is further supported by our cell–cell communication analysis, where we found a higher IGF signaling in aggressive cell populations (Fig. 6D), which is linked to the tyrosine kinase receptor family. Therefore, abruption of IGF signaling using TKIs may provide an effective strategy for targeting aggressive cells in TNBC tumors. Overall, these findings imply that it is potentially possible to repurpose several FDA-approved drugs to overcome chemoresistance and advocate using different drugs pre- and post chemotherapy either alone or in combination.

## Loss of ACTN1 enhances sensitivity to chemotherapy

We were next keen to explore whether any of our 101 signature genes contribute to TNBC chemoresistance through enhanced cell survival and if they can be therapeutically targeted for improving chemotherapy response. Here we decided to make use of cancer dependency maps (DepMap) (Tsherniak et al, 2017) for the first screening followed by additional functional validation (Fig. 8A). To identify the most relevant targets amongst our signature, we prioritized candidate genes by applying a multi-level evidence matrix (Expanded Table EV1) based on (1) high expression in chemoresistant TNBC patients (scRNA-seq, and or bulk RNA-seq); (2) high expression in TNBC tumors and cancer cell lines; (3) high expression in EMT-high TNBC tumors (TCGA) and mammary EMT (mammary epithelial cells), (4) and association of their high expression with reduced survival (relapse-free survival in TNBC and lymph node-positive TNBC tumors). This approach uncovered 12 genes (Fig. EV3B) that qualify the above criteria, including *ACTN1* (encoding α-actinin-1) and *SFRP1* (encoding secreted frizzled-related protein 1) among 20 genes.

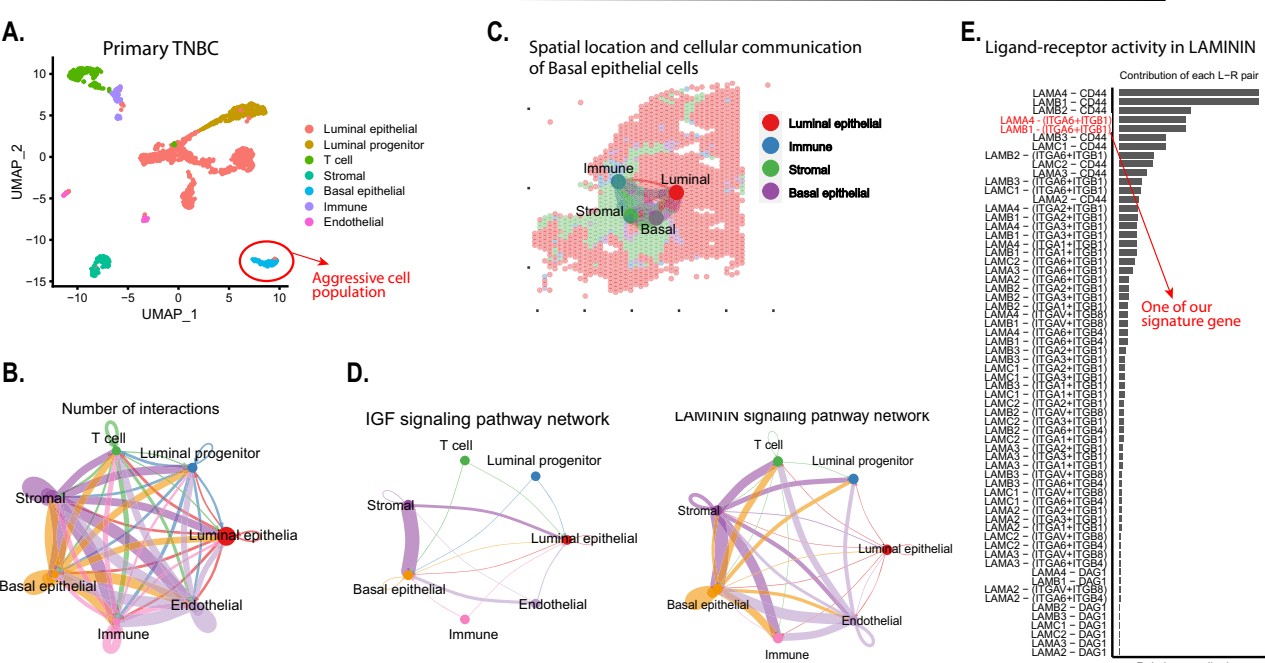

Cell-cell communication network in primary TNBC tumors

**A.** Primary TNBC

**B.** Number of interactions

**C.** Spatial location and cellular communication of Basal epithelial cells

**D.** IGF signaling pathway network / LAMININ signaling pathway network

**E.** Ligand-receptor activity in LAMININ

Cell-cell communication network in post-treated Chemoresistance vs Chemosensitive TNBC tumors

**F.** Chemosensitive cells (Post treated)

**G.** Chemoresistance cells (Post treated)

**H.** Number of interactions

**J.** Signaling pathway activity

**I.** Cell-cell communication network across sensitive and resistant cells

Number of interactions − Sensitive / Number of interactions − Resistant

**K.** Laminin signalling network: Resistant / Laminin signalling network: Sensitive

◄ **Figure 6. Chemoresistant cells communicate through distinct signaling pathways which is enhanced following exposure to chemotherapy.**

(A) UMAP clusters of scRNA-seq datasets of primary TNBC tumors. (B) Circle plot showing number of interactions between the cell types in primary TNBC scRNA-seq dataset. (C) Cell–cell communication of the epithelial cells including basal epithelial with other cell types within TNBC spatial transcriptome dataset. The line and its width represent the strength of the cellular communications between the cell types on the histological section. (D) Circle plot showing key signaling pathways involved in intercellular communications between the basal and other cell types in TNBC. (E) Bar plot showing ligand-receptor interaction of LAMININ signaling pathway for intercellular communication in TNBC. (F, G) UMAP plot of post chemotherapy-treated chemoresistance and chemosensitive cells. (H) Bar plot showing the total number of interactions in the resistant and sensitive group of cells. (I) Circle plot showing the number of interactions between the cell types in chemosensitive and chemoresistant datasets. (J) Plot shows key signaling pathways enriched in chemoresistant and chemosensitive cells. (K) Circle plot showing the number of interactions of the intercellular signaling pathway in chemosensitive and chemoresistant cells. Source data are available online for this figure.

We subjected these 12 genes for further downstream analysis to shortlist key genes for functional validation. To do this, we examined existing datasets from genome-wide CRISPR-Cas9 and RNAi screens of several breast cancer cell lines (Fig. 8A), and found that the candidate gene *ACTN1* was essential in 12 out of 18 TNBC cell lines, suggesting its importance for the viability of TNBC cells (Fig. 8B). Given the enrichment of our signature in basal-like molecular subtypes of TNBC, we selected the HCC1806 (BL1/2 like) and MDAMB-468 (BL2like) cell lines for further investigation, which also expresses sufficient levels of *ACTN1* (Brautigam et al, 2016) (Fig. EV3C).

In parallel, an integrated scRNA-seq analysis of healthy breast (Data ref: Bhat-Nakshatri et al, 2021), primary TNBC (Data ref: Karaayvaz et al, 2018) and chemotreated tumors (Data ref: Kim et al, 2018) (Fig. 8C), revealed that ACTN1 exhibits higher expression in basal cells within TNBC tumors (Fig. 8C,D, left violin plot). Further analysis showed that *ACTN1* was not expressed in healthy breast cells but was upregulated in TNBC primary tumors and further elevated in chemoresistant but not chemosensitive TNBC (Fig. 8D, right violin plot). These observations were confirmed by immunohistochemistry analysis of normal and malignant breast tumor (Fig. 8E).

We further confirmed whether aggressive cells share the origin with basal cell types within the TME that express high levels of ACTN1. To achieve this, we plotted the expression levels of our 20-gene chemoresistance signature across this integrated dataset and found their higher expression in basal epithelial cells (Fig. EV3D) with much higher levels in primary and chemotherapy-treated TNBC tumors. The accuracy of these findings was further confirmed by plotting expression levels of established basal markers such as cytokeratins 5, 14, and 17 and found that the primary TNBC tumors exhibit higher levels of basal markers as compared to healthy breast which is further increased in chemotherapy-treated patients, confirming their basal-like origin (Fig. EV3E, left panel). Notably, this was not the case for luminal epithelial markers and they were enriched in the luminal cluster as expected (Fig. EV3E, right panel).

Furthermore, our spatial transcriptomics analysis revealed that ACTN1 is highly expressed within the basal epithelial cells, (Fig. 8F) which we had found associated with aggressive behavior (Fig. 1F). These striking, multi-level observations clearly vouch for a potential role for ACNT1 in TNBC chemoresistance.

Subsequently, we depleted *ACTN1* gene using small interfering RNAs (siRNA) and treated the knockdown lines with taxane (Paclitaxel), which is the most commonly used chemotherapy agent for TNBC (Fig. 8G). The inhibitory concentration (IC$_{50}$) values of the drugs for the HCC1806 and MDAMB-468 (BL2 like) cell lines were obtained from the GSDC database (Table 1). Here our

approach could deplete ACTN1 with over >90% efficiency, offering an excellent system for functional analysis (Fig. 8H). Interestingly, here we observed a dramatic decrease in cell viability in BL1 subtype (~50%) and a modest but significant reduction in BL2 subtype (~20%) following knockdown of ACNT1 in the presence of Paclitaxel (Fig. 8I). Altogether, these observations identify ACTN1 as an excellent therapeutic target for overcoming TNBC chemoresistance and imply that *ACTN1* levels could be predictive of response to chemotherapy in TNBC patients and its inhibition may re-sensitize chemoresistant cells to chemotherapeutic agents.

## Discussion

Our study has identified subpopulations of basal epithelial cells that associates with chemoresistance and metastasis in TNBC patients. These subpopulations harbor shared transcriptomic profiles of an aggressive disease phenotype across multiple cohorts and defined by robust signature genes. These signature genes are highly enriched in independent cohorts (single-cell and bulk RNA-seq) of pre and post-NAC-treated chemoresistant tumors and associated with activated EMT programs. Earlier studies have shown that aggressive tumors such as TNBC originate from basal-like cells and are often used as a surrogate for identifying the aggressive basal-like breast cancer subtype (Foulkes et al, 2010; Lehmann et al, 2011). Therefore, these observations emphasize that cells with aggressive characteristics could be of basal-like in cell origin. Although EMT has been implicated in chemoresistance and metastasis in breast cancers including TNBC (Kim et al, 2018; Li et al, 2020) (Feng et al, 2021; Wei et al, 2020) and EMT genes were shown to be overexpressed in post-treated chemoresistant tumors (Chen et al, 2019; Kim et al, 2018), the underlying comprehensive transcriptional signatures remained poorly understood, particularly in treatment-naive TNBC patients. Here, we have shown that chemoresistant TNBC cells are defined by the expression of distinct signature genes in treatment-naive and NAC-treated tumors. Using spatial transcriptomics, we further showed that these chemoresistance signature genes are upregulated in basal epithelial cells and spatially reside in close vicinity to stromal compartment in the TNBC tumor microenvironment.

Our analysis shows that elevated levels of our signature genes are associated with significantly reduced survival in TNBC patients, and more closely associated with aggressive TNBC subtypes i.e., basal-like (BL1/2) subtypes (Park et al, 2020; Yin et al, 2020). The poor survival outcome of patients with basal-like tumors is well established, unfortunately without much knowledge of the etiology and underlying transcriptional programs (Hallett et al, 2012; Zhang

**A.**

Clustering of pre- and post- chemo treated groups

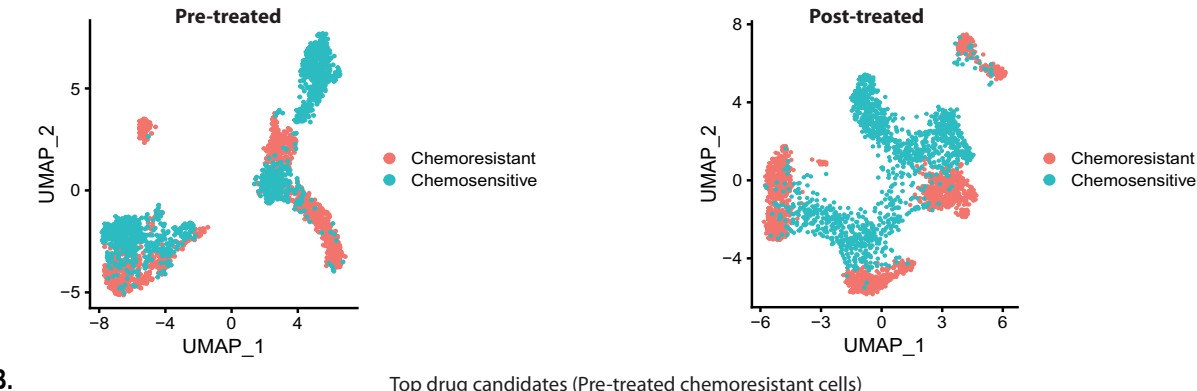

**B.**

Top drug candidates (Pre-treated chemoresistant cells)

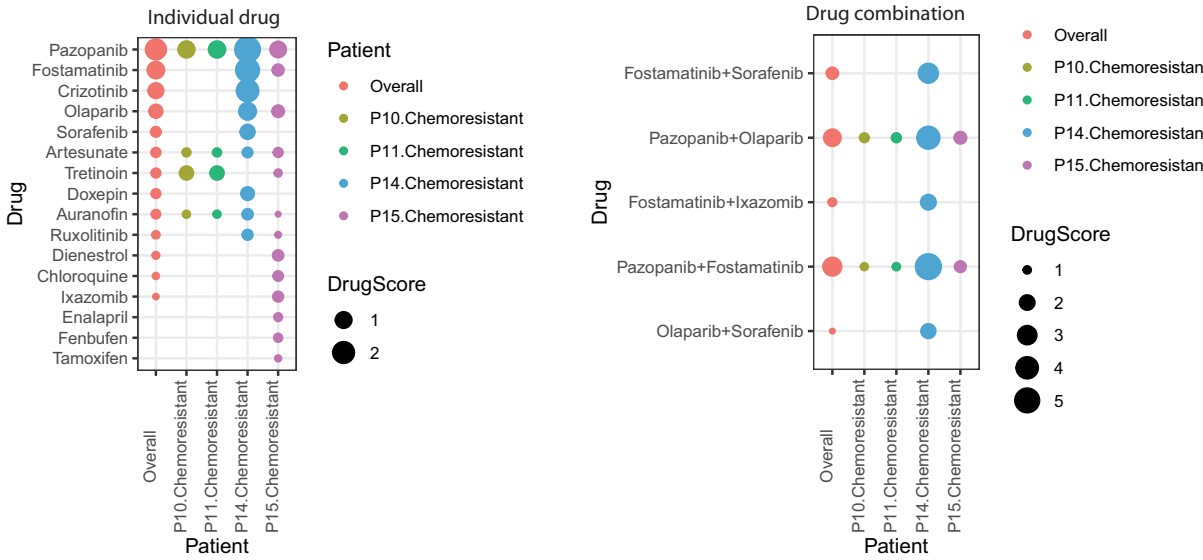

Top drug candidates (Post-treated chemoresistant cells)

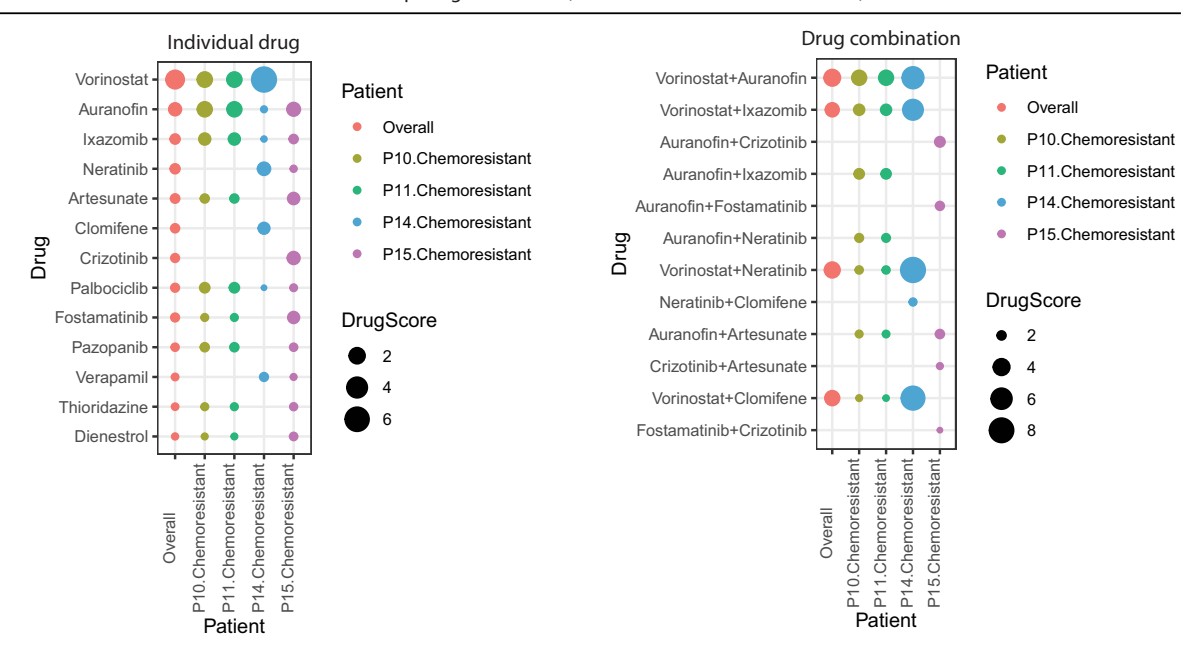

◀ **Figure 7. Drug repurposing analysis of chemoresistant cells identifies potential FDA-approved drugs to overcome chemoresistance.**

(A) UMAP showing cell clusters of pre- and post-NAC-treated TNBC tumors. The single-cell RNA-seq datasets used in Fig. 2 were utilized for drug repurposing analysis using ASGARD pipeline. (B) The top and bottom dot plot show top-ranked drugs (FDR <0.05) that had high drug scores for pre- and post-NAC-treated resistant cells, respectively, for all patients combined (overall) as well as in each chemoresistant TNBC patient. Source data are available online for this figure.

et al, 2020). Furthermore, basal-like cells are known to display cell plasticity in many tumors (Adriance et al, 2005; Gudjonsson et al, 2005), particularly metaplastic tumors with EMT phenotypes (Tan and Ellis, 2013), and associated with metastatic spread (Sarrio et al, 2008) (Alluri and Newman, 2014; Bertucci et al, 2012). However, genes that are known to predict metastasis in TNBC are limited (Qian et al, 2017; Savci-Heijink et al, 2016), which is related to the lack of prognostic indicators. Therefore, we propose that our signature could offer an excellent prognostic indicator in TNBC patients.

Furthermore, these signature genes showed a frequent gain in copy number, compared to other alterations in TNBC patients. This finding suggests that their increased mRNA level in TNBC may be governed primarily through an increase in copy number, which is in line with earlier observations that a gain in copy numbers and acquired mutations contribute to chemoresistance in TNBC patients (Almendro et al, 2014; Balko et al, 2016; Navin, 2014). In addition, we found that our signature genes were elevated in patients with homologous recombination deficiency (HRD) subtype which mostly constitute basal-like characteristics. HR deficiency is known to contribute variably to standard NAC in TNBC patients (Telli et al, 2018). We propose that our signature could infer tumors representing the HRD subtype with basal-like characteristics. These observations also open doors for exploring alternative treatment options, such as PARP inhibitors and other DNA-damaging agents, for these patients.

Although many gene signatures have been reported to predict chemotherapy response in TNBC, their poor accuracy precludes their use in the clinic. Our multi-gene signature ($n = 20$) meets this critical unmet need and can predict a patient's response to NAC with over 90% accuracy in treatment-naive groups and out-performed published gene signatures (Fournier et al, 2019; Juul et al, 2010; Lim et al, 2020; Stover et al, 2016; Witkiewicz et al, 2014). A key reason for this robustness may stem from the origin of our gene signature as it was derived from subpopulation of basal epithelial type using a single-cell transcriptomics approach as compared to all previously published signatures which were based on bulk methods. This clearly highlights the power of single-cell transcriptomics which has emerged as a powerful tool in revealing tumor heterogeneity and thereby in discovering potent biomarkers (Goswami et al, 2020; Rendeiro et al, 2020), including therapy resistance signatures (Fan et al, 2023; Song et al, 2022). Another factor that may account for the superiority of our signature is the underlying biology i.e., EMT represented by our signature genes. Contrary to this, all previous signatures were derived from tumors associated with very different biological functions (Fournier et al, 2019; Juul et al, 2010; Lim et al, 2020; Stover et al, 2016; Witkiewicz et al, 2014). Therefore, the ability of our signature to predict which patients will have pCR or RD using our 20-gene panel will help clinicians make informed decisions prior to chemotherapy and explore alternative therapeutic strategies.

The interaction of tumor stroma with other cells within the TME has been implicated in therapeutic resistance across multiple cancers (Ni et al, 2021; van der Spek et al, 2020), including breast cancer (Mao et al, 2013; Mao et al, 2015) and targeting these interactions in TNBC was shown to be able to sensitize tumors to docetaxel chemotherapy in mouse models (Cazet et al, 2018). Our cell–cell communication analysis revealed that compared to sensitive cells, aggressive cells engage in much stronger intercellular communication with cells in the close vicinity to stroma using distinct signaling pathways, such as IGF and LAMININ cascades, both before and after exposure to chemotherapy. Interestingly, these pathways utilize one of our signature genes ITGB1, implying that a higher expression of ITGB1 could result in increased intracellular communication through these pathways and mediate chemoresistance in TNBC. In line with our observations, LAMININ signaling has been implicated in tumor progression and EMT by promoting the breakdown of collagen IV (α1) to facilitate migration (Liu et al, 2018; Zeisberg and Neilson, 2009) and a few reports have proposed that LAMININ signaling regulates the resistance of cancer cells to therapeutic agents (Weaver et al, 2002; Yang et al, 2010).

Given no clear alternate therapy options for TNBC, our drug repurposing analysis suggested that treatment-naive patients may benefit from TKIs, whereas post-NAC-treated resistant tumors may have sensitivity to HDAC inhibitor (Vorinostat), DMARDs (Auranofin), proteasome inhibitor (Ixazomib) and TKIs (Neratinib, Crizotinib, Fostamatinib and Pazopanib) as individual medications or in combination. For example, Auranofin is DMARDs inhibitor is originally used to treat rheumatoid arthritis but has shown potential in lung cancer (Hou et al, 2018). Similarly, Artesunate is an anti-malarial drug that is effective against leukemia, breast cancer, gastrointestinal tumors and other types of cancers (Khanal, 2021; Mancuso et al, 2021; Pirali et al, 2020). Likewise, Pazopanib and Vorinostat are being used to treat advanced renal cell carcinoma and cutaneous T-cell lymphoma, respectively. These encouraging observations have opened the door for future experimental and clinical investigation of these drugs in the treatment of residual TNBC tumors.

Our further functional analysis revealed that a high expression of one of our signature gene *ACTN1* confers survival advantage to TNBC cells and its depletion enhances the sensitivity to taxane-based chemotherapies in TNBC cells. These findings are in line with recent reports showing that depletion of *ACTN1* (Chen et al, 2021; Xie et al, 2020) inhibited the growth of hepatocellular carcinoma and oral squamous cell carcinoma, however it has not yet been explored for improving sensitivity to NAC in TNBC. ACTN1 is an actin-binding cytoskeletal protein that is frequently overexpressed in human breast cancers and linked with the poor prognosis in basal-like breast cancer (Kovac et al, 2018). Its increased mRNA levels result in the destabilization of E-cadherin-based adhesions, which can contribute to cancer progression

**A.**

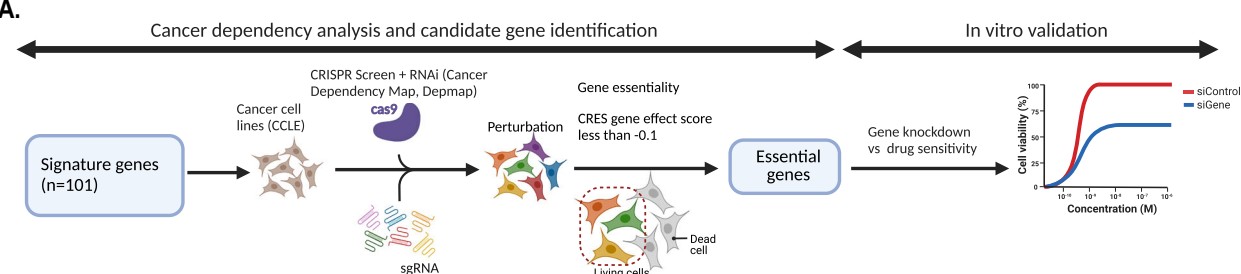

**B.**

Genetic dependency map of ACTN1 across TNBC lines

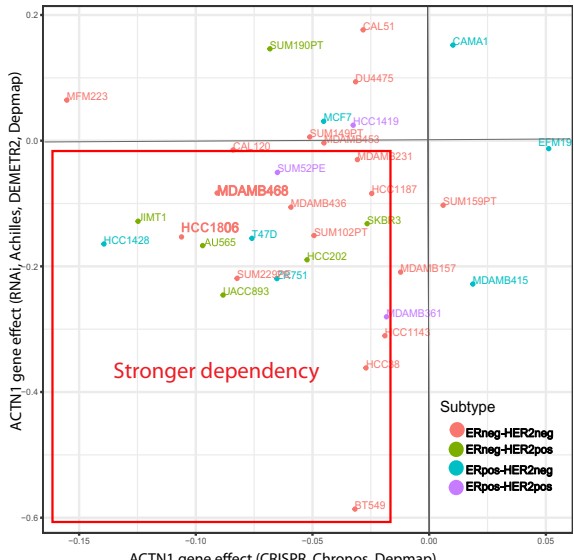

**C.**

Integrated scRNA-seq datasets of Healthy breast, primary TNBC and chemosensitive and chemoresistant TNBC

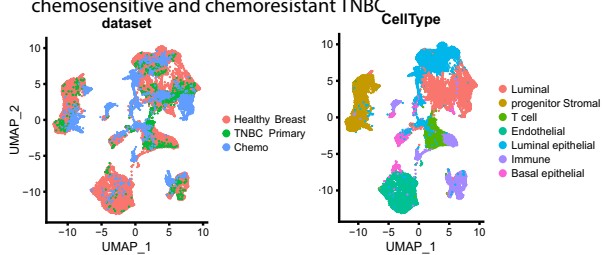

**D.**

Expression level of ACTN1 in celltype of Healthy breast, primary TNBC and chemosensitive and chemoresistant TNBC

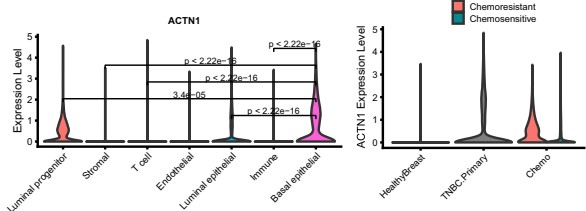

**E.**

IHC staining of ACTN1 in normal breast and breast cancer

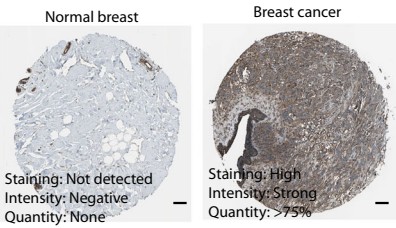

**G.**

Effect of ACTN1 knockdown on cell viablity and drug sensitivty

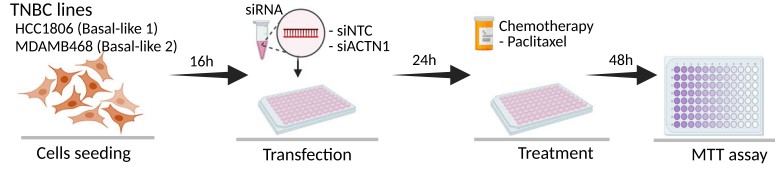

**F.**

Expression of ACTN1 in TNBC spatial transcriptome

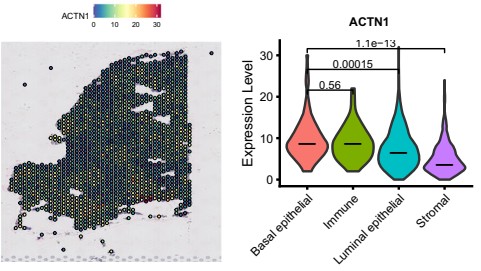

**H.**

ACTN1 knockdown efficiency (n=2 replicates)

**I.**

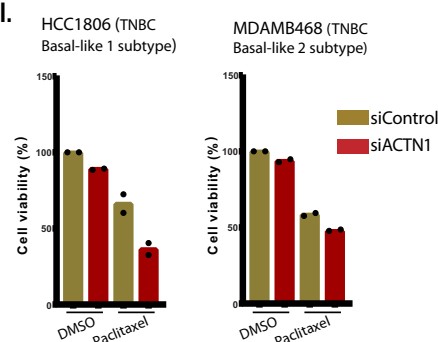

◀ **Figure 8. The cancer dependency map of our signature genes and its correlation with drug sensitivity in TNBC lines.**

(A) The workflow of cancer dependency analysis and candidate gene validation performed in the present study. (B) The scatter plot shows the genetic dependency of ACTN1 in TNBC lines. The X and Y axis indicate dependency score derived from CRISPR and RNAi-based perturbation screens, respectively. The dependency score was obtained from CRISPR (CRISPR, Chronos, Depmap) and RNAi (Achilles, DEMETR2) from DeepMap (https://depmap.org/portal/download/). (C) UMAP plot is the integrated scRNA-seq profiles of healthy breast, primary TNBC and pre-post chemotherapy-resistant and -sensitive cells. (D) The expression of the *ACTN1* gene was plotted in the violin plot, showing mRNA levels across different cell types of integrated healthy breast, primary TNBC and chemotherapy-treated TNBC cells. The right violin plot shows expression of ACTN1 across healthy, primary TNBC and chemotreated-resistant and -sensitive TNBC cells. The significance test of expression levels of ACTN1 between the cell types was performed using two-tailed unpaired Wilcoxon test in stat_compare_means() of ggpubr package. (E) IHC staining (scale bar, 10 μm) analysis of protein levels of ACTN1 in samples of healthy breast and breast cancer, derived from human protein atlas (HPA). (F) ACTN1 expression level in a TNBC spatial transcriptome dataset. The left plot shows the expression of ACTN1 in the spatial transcriptome dataset. The right plot shows the enrichment of the ACTN1 gene in spatially annotated cell types (deconvoluted from the primary TNBC scRNA-seq dataset used in Fig. 1B). The significance test of expression levels of ACTN1 between the cell types was performed using two-tailed unpaired Wilcoxon test in stat_compare_means() of ggpubr package. (G) Candidate gene ACTN1 knockdown and drug-sensitivity workflow. (H) Barplot shows knockdown efficiency of ACTN1 in HCC1806 and MDAMB-468 cell line. The data with error bars are shown as mean ± SD. Statistical significance was calculated by two-tailed Student t test. (I) Barplot shows cell viability after transfection with the indicated siRNAs of control (siControl) and candidate genes (siACTN1) knockdown in combination with Paclitaxel chemotherapy in micromolar concentration (μM). Data are representative of two experiments conducted in triplicates. The dot on each bar shows individual data points of two experiments. Source data are available online for this figure.

**Table 1. The inhibitory concentration (IC$_{50}$) values of the drugs.**

| Cell lines | Paclitaxel (μM) |
| --- | --- |
| HCC1806 | 0.013 |
| MDAMB-468 | 0.016 |

The IC$_{50}$ values of the drugs for cell lines were obtained from Genomics of Drug Sensitivity in Cancer (GSDC). The IC$_{50}$ is represented in micromolar (μM) concentration.

through partial EMT (Kovac et al, 2018). Similarly, study showed targeting ACTN1 by Oroxylin A could remodel stromal microenvironment and restrain breast cancer metastasis (Cao et al, 2020). Furthermore, ample existing and growing evidence has shown that structural proteins can be excellent targets for cancer therapy (Dumontet and Jordan, 2010; Jordan and Wilson, 2004; Trendowski, 2014), including the widely established chemotherapeutic agent Paclitaxel which functions via targeting microtubules (Dumontet and Jordan, 2010; Trendowski, 2014). Notably, several other drugs that function via similar mechanisms are currently in the clinical trial (Chen et al, 2014; McGough et al, 1998; Rao et al, 2019; Suzuki et al, 2010). Therefore, our study provides the first evidence that loss of ACNT1 can improve the chemotherapy response in TNBC cells and hence warrants further investigation to establish its clinical utility. Together with our data, these observations support a critical role for ACTN1 in conferring chemoresistance in tumor cells, and its targeting holds potential to re-sensitize tumors to taxane-based chemotherapeutics agent, which are still standard care of treatment regimens for TNBC patients (Mustacchi and De Laurentiis, 2015).

While our extensive investigation using several state-of-the-art omics approaches and independent validation in large patient datasets has shed light on the molecular drivers of chemoresistance in TNBC, the current study does have some limitations. This mainly relates to a relatively small sample size for single-cell RNA-seq and spatial transcriptomics analysis, current limitations in computational methodologies including the detection of a limited set of genes in single-cell and spatial transcriptomics analysis as well as dependency on current cell-type annotations. Furthermore, the discovery power could be enhanced by an integration of multimodal clinical features with molecular signatures. In addition, future studies should investigate tumor samples derived from different ethnic groups across various geographical regions to ensure a wider applicability of our findings.

Overall, our integrated analysis of multimodal data from several independent cohorts has uncovered key subpopulations, their underlying genes signatures, and potential mechanisms involved in TNBC chemoresistance. These observations will not only facilitate therapy management by providing an upfront prediction of therapy response but also create opportunities for novel therapeutic intervention including drug repurposing for better clinical management of TNBC, and thus could potentially improve the survival and quality of life for these patients.

## Methods

### Identification of chemoresistance and metastasis cell populations using single-cell transcriptome profiling of TNBC tumors

Our first goal was to uncover cellular diversity in TNBC and assess the presence of cells exhibiting chemoresistance and metastasis-like features (Fig. 1A). For this, we analyzed the single-cell RNA-seq profile of nearly 9000 cells (8974 cells) derived from 33 TNBC patients, including patients treated with neoadjuvant chemotherapy (NAC) from four different studies (Data ref: Chung et al, 2017; Data ref: Gulati et al, 2020; Data ref: Karaayvaz et al, 2018; Data ref: Kim et al, 2018). All four datasets were analyzed with the uniform parameters in the Seurat package (Version 4.0.1), where cell pre-processing was done, and cells were removed having unique feature counts >2500 or <200 and had mitochondrial reads >5%. Further downstream analysis was performed using the same Seurat package, in which data normalization was performed, followed by variable feature selection, dimensionality reduction, and clustering. The clusters were identified with the UMAP reduction method. For assigning cell-type identity to each cluster, we used manually curated marker lists of breast cell types from the CellMarker database (Zhang et al, 2019). Batch effect across multiple samples were regress out and the integration of scRNA-seq datasets were done using canonical correlation analysis (CCA) method in Seurat. The authenticity of batch correction was evaluated based on the fact that cells are clustered based on the distinct cell type and not based on patient samples.

## Identification of disease aggressiveness associated (metastasis and chemoresistance) subpopulations

As single-cell analysis has provided a better resolution to cell populations associated with poor clinical outcomes, we analyzed cell subpopulations for the enrichment of metastasis and treatment response associated with distinct gene expression signatures. For this, we used 49 metastasis signature genes identified in patient-derived murine xenograft models of TNBC and which stratify high versus low metastatic burden tumors (Lawson et al, 2015). For therapy resistance signature genes, we used 143 genes (from Cluster AHI) which were earlier found to be highly correlated with high Ki-67 expression and achieved the highest chemotherapy resistance scores in basal-like breast cancer subtypes (Balko et al, 2012).

## Copy number variation analysis

In order to identify malignant cells within the primary TNBC scRNA-seq datasets, we used infercnv (v1.3.3) package (Patel et al, 2014). We used $n = 240$ normal mammary epithelial cells as reference to identify somatic copy number variations (CNV) within TNBC epithelial cell populations, including basal epithelial cells. Each cell was further computed score as "InferCNV.score" for the extent of CNV signal, based on the number of genes with copy number alterations (CNA) within the cells, computed by infercnv_obj@expr.data output in inferCNV. Potential malignant cells were identified based on the CNV signal (InferCNV.score) higher than 0.2 compared to normal epithelial cells which scored less than 0.2.

## Functional analysis of identified subpopulation and reproducible signature gene identification

The set of genes that shows distinct expression in the identified cluster was compared between all three scRNA-seq datasets to identify reproducible signature genes across the same cell types in different independent studies. These reproducible genes were further analyses for biological functional analysis using the ShinyGO gene ontology enrichment analysis tool. Functional pathway activity in single-cell pre and post-NAC-treated datasets was performed using hallmark gene signature derived from Molecular Signatures Database (MSigDB) integrated in escape R package (version 1.12.0) (Borcherding et al, 2023). The expression profiling of signature genes was investigated across TCGA breast cancer subtypes to ensure these signature genes are TNBC specific. For TNBC molecular subtype analysis, we retrieved RNA-seq expression levels from METABARIC (Data ref: Curtis et al, 2012), and TCGA-BRCA (Data ref: Cancer Genome Atlas Network, 2012) dataset. The TNBC molecular subtyping information of the METABARIC, and TCGA-BRCA cohorts was derived from the recently published study (Lehmann et al, 2021). In addition, these genes were also screened in large breast cancer cohorts in EMT-High and EMT-Low groups. For this, we first retrieved the TCGA mRNA expression z-scores of breast cancer tumors from cBioPortal (Data ref: Gao et al, 2013) using the "cgdsr" R package (version 1.3.0) and extracted TNBC tumors based on the ER,PR and HER2 status. Next, the EMT scores were calculated for each sample by subtracting the average expression z-scores of the 4 "epithelial"

markers (*ESRP2*, *OVOL1*, *OVOL2*, and *CLDN3*) from the average expression z-scores of the 6 mesenchymal' markers (*ZEB2*, *SNAI2*, *TWIST1*, *TWIST2*, *VIM*, and *FN1*). The tumor samples were then classified as EMT-High (defined by EMT scores ≥ highest 1/3) and EMT-Low (defined by EMT scores ≤ lowest 1/3) based on the calculated EMT scores (Lou et al, 2016). The "Pheatmap" R package (version 1.0.12) was used to construct the heatmap of expression levels of the 10 markers in the EMT-High and EMT-Low groups defined above. Finally, the average expression profile of identified signature genes in these EMT group's breast cancer cohorts was plotted using the ggplot2 package.

## Development of RNA classifiers for predicting pathological complete response (pCR) or residual disease (RD)

For developing predictive models for stratifying TNBC tumors into pathological complete response (pCR) or residual disease (RD) groups, we used five publicly available microarray-based gene expression datasets GSE25055 (Data ref: Hatzis et al, 2011), GSE25065 (Data ref: Hatzis et al, 2011), GSE20194 (Data ref: Shi et al, 2010), GSE41998 (Data ref: Horak et al, 2013) and GSE20271 (Data ref: Tabchy et al, 2010). We extracted normalized expression levels of 101 signature genes from two studies GSE25055 and GSE25065 of 177 patients to train the model. For testing the performance of the model we used 269 TNBC patients from the remaining three datasets GSE20271, GSE41998, and GSE20194. The treatment details of all the patients were available and hence used as a class for binary classification. The clinical details of each dataset were shown in the Fig. 5B. The GSE25055 and GSE25065 datasets include 177 TNBC patients enrolled in an NAC trial who received sequential or combination of Taxane (paclitaxel, 12 weekly cycles) and anthracycline-based regimens (4 cycles of doxorubicin or epirubicin, fluorouracil, and cyclophosphamide [FAC/FEC]; 4 cycles of doxorubicin and cyclophosphamide [AC]). The GSE20271 dataset includes 59 TNBC patients in an NAC trial who received either weekly Taxane (12 cycles of paclitaxel) followed by four cycles of 5-fluorouracil, anthracycline (doxorubicin) and cyclophosphamide (T/FAC) or FAC. The GSE20194 includes 71 newly diagnosed TNBC patients who received 6 months of NAC including Taxane (12 cycles of paclitaxel), followed by four cycles of 5-fluorouracil, cyclophosphamide and anthracycline (doxorubicin) (TFAC). The GSE41998 dataset consists of 139 patients from a randomized, open-label, multicenter, phase II trial (NCT00455533) enrolled previously untreated women with histologically confirmed primary invasive breast adenocarcinoma. Patients received sequential neoadjuvant therapy starting with four cycles of AC (doxorubicin 60 mg/m² intravenously and cyclophosphamide 600 mg/m² intravenously) given every 3 weeks, followed by 1:1 randomization to either ixabepilone (40 mg/m2 3-h infusion) every 3 weeks for 4 cycles, or paclitaxel (80 mg/m² 1-h infusion) weekly for 12 weeks.

We used Lasso and Elastic-Net Regularized Generalized Linear Models in glmnet for best fitting the model with tenfold cross-validation to remove bias. In this, we first ranked the features to select the minimum set of features with maximum predictive power for pCR and RD. For comparing the performance of the 20-gene model with other published signature genes in stratifying TNBC patients of responsive and non-responsive groups, we utilized gene signatures from five earlier studies (Fournier et al, 2019; Juul et al,

2010; Lim et al, 2020; Stover et al, 2016; Witkiewicz et al, 2014). The first signature was derived from the Witkiewicz et al, study which consists of nine genes that exhibited strong predictive power in classifying pCR and residual patients against TA, TFAC, and FAC chemotherapy response. The second signature was a stromal-related sixteen signature that reflects the activation state of the tumor stroma and predicts poor response to anthracycline-based neoadjuvant chemotherapy in TNBC patients. The Jull et al, signature consists of six mitotic genes that were a strong predictor of pCR against neoadjuvant paclitaxel chemotherapeutics agents. The Stover et al, an eighteen-gene combined proliferation-immune "meta-signature" was a strong predictor of response to neoadjuvant chemotherapy in TNBC patients. Finally, Lim et al, the study identified fifteen immune-related gene signatures using NanoString nCounter Immunology Panel mRNA expression quantification platform and found was strongly associated with pCR in TNBC patients.

## Genetic dependency analysis and drug-sensitivity analysis

The genetic dependency of our signature genes on cancer cells was investigated using Genome-wide CRISPR dependency and RNAi screen data available from Depmap portal https://depmap.org/portal/download/. The Gene effect scores for our signature genes were downloaded from the CRISPR, Chronos, Depmap and RNAi, Achilles, DEMETR2 for more than 1300 cancer cell lines including 18 TNBC lines available from Cancer Cell Line Encyclopedia (CCLE). A lower score indicates a higher likelihood that the gene is essential for the cell line. A score of 0 indicates no essentiality, while a score of negative value is comparable with the median of all pan-essential genes, i.e., the genes which are essential for every cell line.

## Cell–cell communication analysis

Cell–cell interaction analysis was applied on our scRNA-seq dataset of both primary TNBC (Data ref: Karaayvaz et al, 2018), and chemoresistant tumors (Data ref: Kim et al, 2018), using CellChat algorithm (Jin et al, 2021). We applied CellChat with default settings with 1000 permutations, which quantitatively characterizes and compares the inferred cell–cell communication based on the average expression of the ligands and receptors in various cell populations.

## Drug repurposing analysis

The drug repurposing analysis was performed on post-treated chemoresistance cells (Data ref: Kim et al, 2018) to identify potential drug candidates using ASGARD tool (He et al, 2023), which ranks FDA-approved drugs against cell populations using scRNA-seq datasets. ASGARD uses the differentially expressed genes (in our case, between resistant and sensitive) as inputs to identify drugs that can significantly (single-cluster FDR <0.05) reverse their expression levels in the L1000 drug response dataset.

## Cell culture

The TNBC lines HCC1806 were maintained in RPMI 1640 (Gibco, 21875034) medium supplemented with 10% FBS, 1% glucose, 1 mM

sodium pyruvate (Thermo, 11360070). MDAMB-468 cells were maintained in DMEM (Dulbecco's modified Eagle's medium) with 10% FBS. Cells were grown as monolayers at 37 °C in humidified $CO_2$ (5%) incubator. Both cell lines were originally obtained from ATCC and tested negative for mycoplasma prior to the study.

## siRNA transfection

The scrambled siRNA control and ON-TARGETplus SMARTpool siRNA targeting human ACTN1 were purchased from Dharmacon. Transfection was performed using Lipofectamine™ RNAiMAX (Invitrogen, 13778150) according to the manufacturer's instructions. In brief, cells were seeded at 12,000 cells per well the day before the transfection. siRNA at a final concentration of 5 pmol was diluted in 25 µL of Opti-MEM and 1.5 µl of Lipofectamine RNAiMAX was diluted in 25 µl of OPTI-MEM. The diluted siRNA and Lipofectamine RNAiMAX were mixed and incubated at room temperature for 10 min. Ten microliters of transfection mixture were added to each well of 96-well plates. Twenty-four hours later, the transfection cocktail was replaced with complete DMEM. The primer sequences of ACTN1 and GAPDH is shown in Table 2.

## MTT assay and drug-sensitivity analysis

Transfected cells were cultured for 24 h and treated with paclitaxel at the desired concentration for each cell line (Table 1). DMSO served as vehicle control. The treated cells were incubated for 48 h and a cytotoxicity assay was performed using an MTT assay kit (Roche, 11465007001) according to the manufacturer protocol. Briefly, 10 µl MTT (5 mg/ml) was added to each well and allowed to form formazan crystals for four hours in the incubator. In total, 100 µl of solubilization solution was added to each well and incubated overnight in the incubator in a humidified atmosphere. The next day, complete solubilization of the purple formazan crystals was confirmed, and then the absorbance values were determined using a microplate reader (BMG FLUOstar Omega) at 590 nm. The experiments were repeated twice, and data are represented as mean ± SD from three technical replicas. No blinding was done.

## Statistical analysis

GraphPad Prism (v 9.2; GraphPad Software, San Diego, CA, USA) was used for statistical analyses. All experiments were performed in two independent biological replicates, each comprising three technical replicates, of which details can be found in respective figure legends. The statistical analysis was performed using two-tailed Student's $t$ test to measure significant differences between the groups (for $n = 2$ groups). A $P$ value of <0.05 was considered statistically significant.

The statistical analysis of publicly available datasets was performed using two-tailed Wilcoxon test (for $n = 2$ groups) or

Table 2. Primer sequences used in Fig. 8H.

| Gene | Primer name | Sequence |
|---|---|---|
| ACTN1 | VTQUB 405_ACTN1_FP | CAAGATCTCCAACGTCAACAAG |
|  | VTQUB 406_ACTN1_RP | CACATTCCCATCCACGATTTC |
| GAPDH | VTQUB 572_GAPDH_FP | GAAGTATGACAACAGCCTCAAG |
|  | VTQUB 573_GAPDH_RP | CGATACCAAAGTTGTCATGGAT |

**The paper explained**

**Problem**

Triple-negative breast cancer (TNBC) is the most aggressive type of breast cancer and is hard to treat. It spreads fast, does not respond well to chemotherapy, and often leads to poor outcomes in patients. Despite advances in the field, the molecular basis of these aggressive behaviors remains poorly understood.

**Results**

In this study, we used advanced techniques that allow a closer look at the TNBC tumor cells and their genes at single-cell and spatial resolution. This analysis identified specific groups of cells in the tumor that exhibit resistance to chemotherapy. Furthermore, these cells express certain genes that are highly active and predictive of future response to chemotherapy. Interestingly, high levels of ITGB1 improve cell communication, and ACTN1 expression gives cells a survival advantage, fostering resistance to chemotherapy. Furthermore, we identified existing drugs that may be repurposed against chemoresistant tumors.

**Impact**

Our findings provide an explanation on why certain TNBC tumors are resistant to chemotherapy and proposes a biomarker for predicting patient's response to chemotherapy. This work opens avenues for precision medicine, providing stratification biomarker and alternative therapies for better managing TNBC patients resistant to traditional chemotherapy.

Kruskal–Wallis test (for $n > 2$ groups) using stat_compare_means() of ggpubr package. A $P$ value of $<0.05$ was considered statistically significant. The statistical testing details is also included in the figure legend of the figures.

## Data availability

Primary datasets generated in this study: The RNA-seq datasets derived from the biological triplicates of TGFβ-treatment time course of HMLE cells is submitted to the Expression Omnibus database under the accession code GSE252315.

## Peer review information

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

## Acknowledgements

The authors would also like to thank the members of the Tiwari lab for their cooperation and critical feedback throughout this study. The support from the Core Facilities of the Queen's University Belfast is gratefully acknowledged. This study was supported by the Deutsche Forschungsgemeinschaft TI 799/1-3 and Innovation to Commercialisation of University Research (ICURe).

## Author contributions

**Mohammed Inayatullah**: Conceptualization; Resources; Data curation; Software; Formal analysis; Investigation; Methodology; Writing—original draft; Project administration. **Arun Mahesh**: Investigation. **Arran K Turnbul**: Investigation. **J Michael Dixon**: Investigation. **Rachael Natrajan**: Investigation. **Vijay K Tiwari**: Conceptualization; Resources; Formal analysis; Supervision; Funding acquisition; Project administration; Writing—review and editing.

## Disclosure and competing interests statement

The identified 20-gene biomarker panel has been filed for a patent for the prediction of chemotherapy response in triple-negative breast cancer (TNBC) with accession WO 2023/099890 A1; PCT/GB2022/053035.

# Expanded View Figures

**Figure EV1.** (A) scRNA-seq data analysis of pre-treated primary TNBC patients identified a similar subpopulation in two independent scRNA-seq of TNBC datasets. The cell clusters marked under lines are representing, cells belonging to the breast cancer subtype. The genes defining each cluster were annotated against the cellmarker database and cell-type identities were assigned to each cluster. (B) Expression of metastasis-associated genes in pre-treated TNBC patients scRNA-seq datasets. The metastasis signature of 49 genes was used by Lawson et al, 2015 and their average expression was plotted across each cluster in both datasets. (C) Chemoresistance signature of 143 genes was used by Balko et al, 2012 and their average expression was plotted across each cell types in both datasets. (B, C) The significance test of expression levels of metastasis and chemoresistance genes between the cell types was performed using two-tailed unpaired Wilcoxon test in stat_compare_means() of ggpubr package. (D) The violin plots show expression of malignant cell markers of Luminal and basal breast cancer type. The cancer cell marker of basal and luminal epithelial type was retrieved from CellMarker database and plotted on our primary TNBC dataset. (E) Spatial transcriptome data of two recurrent TNBC patients. The left plots show the H&E staining (scale bar, 10 μm) of two TNBC tumors. The middle plot shows the spatial location of basal epithelial cells within these spatial datasets. Right plot showing the mean expression of our 101 signature genes in these spatial transcriptome datasets.



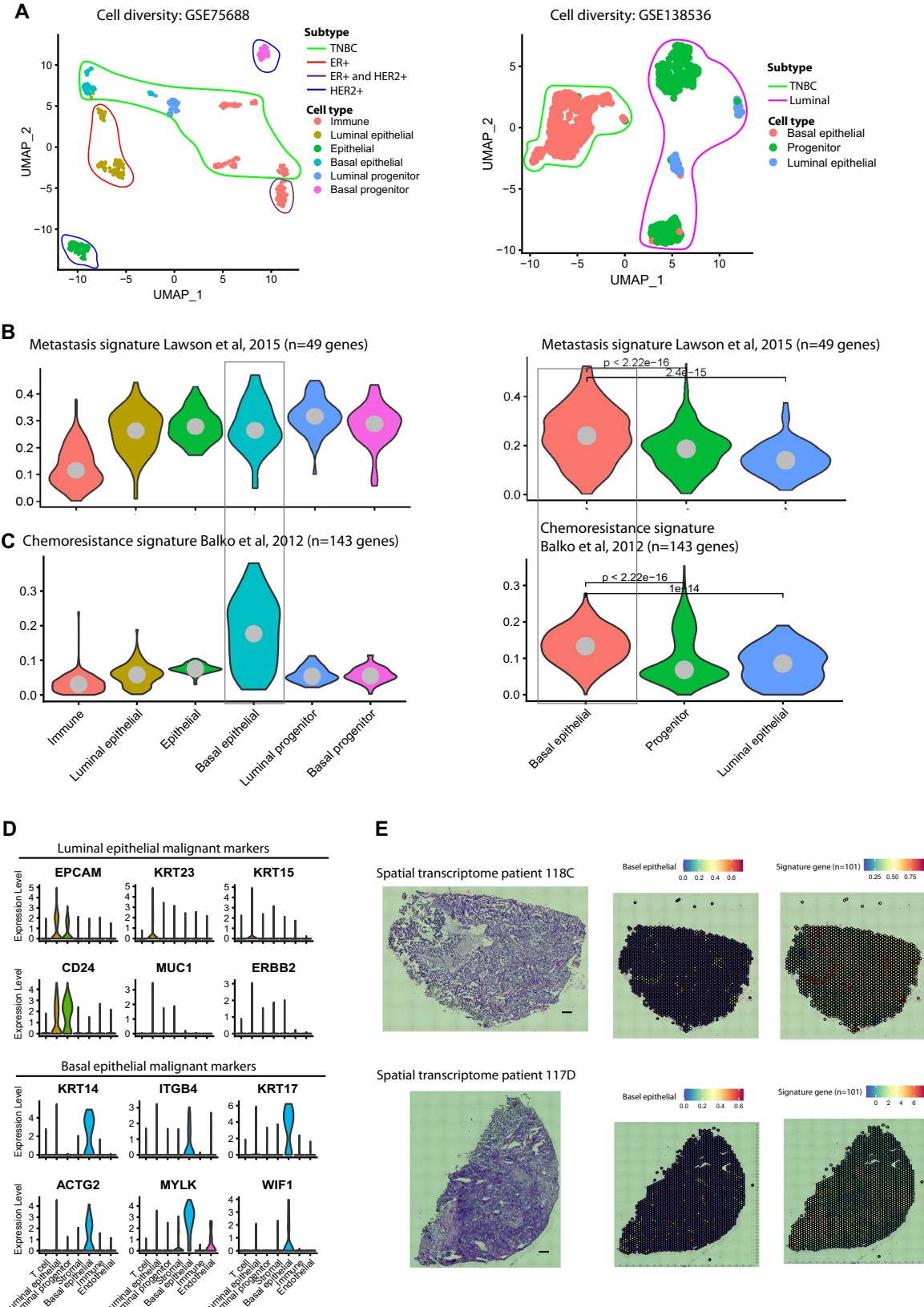

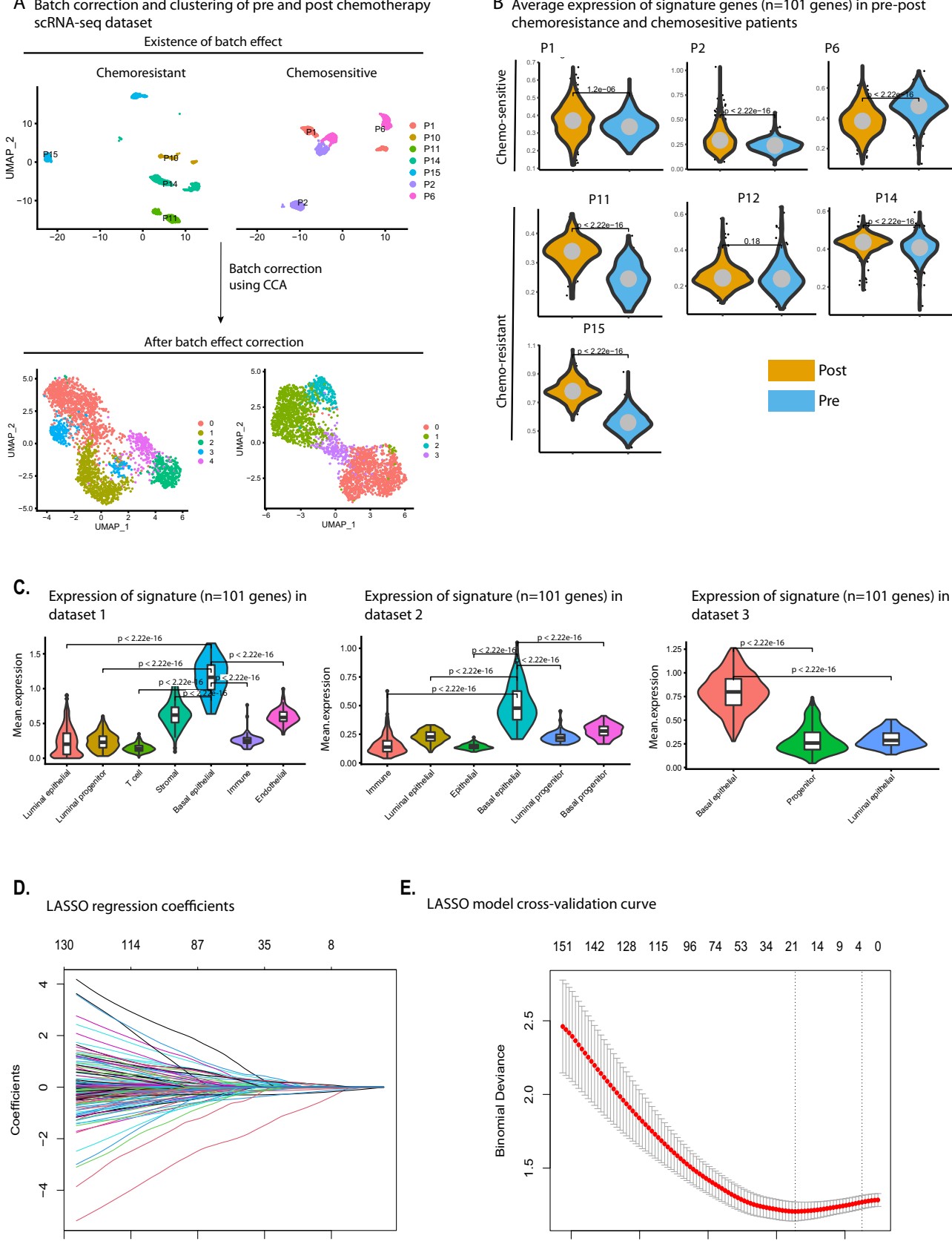

A  Batch correction and clustering of pre and post chemotherapy scRNA-seq dataset

B  Average expression of signature genes (n=101 genes) in pre-post chemoresistance and chemosesitive patients

C.  Expression of signature (n=101 genes) in dataset 1

Expression of signature (n=101 genes) in dataset 2

Expression of signature (n=101 genes) in dataset 3

D.  LASSO regression coefficients

E.  LASSO model cross-validation curve

◀ **Figure EV2.** (A) The upper umap plot shows existence of possible batch effect in resistant and sensitive datasets. The batch effects regress out using canonical correlation analysis (CCA) and samples were integrated. The bottom umap plots shows removal of possible batch effects from the datasets. Total cluster identified in the single-cell datasets of 7 TNBC patients pre- and post chemotherapy are also shown in the same bottom plot. (B) The average expression profile of signature genes in each patient of pre- and post-treated groups. The significance testing of expression of signature genes between the chemotherapy-treated and untreated groups was performed using two-tailed unpaired Wilcoxon test in stat_compare_means() of ggpubr package. (C) Violin plot showing average expression of signature genes across clusters of all three primary TNBC tumor datasets. The average expression of all 101 signature genes was plotted in all three primary TNBC scRNA-seq datasets and confirmed their activation in similar subpopulations of basal epithelial cells. In the box-and-whisker within violin plots, the horizontal lines mark the median, the box limits indicate the 25th and 75th percentiles, and the whiskers extend to 1.5× the interquartile range from the 25th and 75th percentiles. The significance test of expression levels of signature genes between the cell types was performed using two-tailed unpaired Wilcoxon test in stat_compare_means() of ggpubr package. (D) The coefficients from the Lasso fit represent the contributions of the 20 genes expression in the model. The plot shows lasso regression coefficient values in which each curve corresponds to a variable. It shows the path of its coefficient against the Log Lamda of the whole coefficient vector as λ varies. The axis above indicates the number of nonzero coefficients at the current λ, which is the effective degrees of freedom (df) for the lasso. (E) The selection of tuning parameter (λ) in the LASSO model based on the tenfold cross-validation. The plots are showing a cross-validation curve (red dotted line) along with mean binomial deviance against a range of Log(λ). The vertical dotted lines represent lambda. min and lambda.1se. This panel shows the changes in partial likelihood deviance with λ values. The 20 genes were selected according to the most regularized model such that the error is within one standard error of the minimum.

**A.**

Ligand-receptor activity in LAMININ

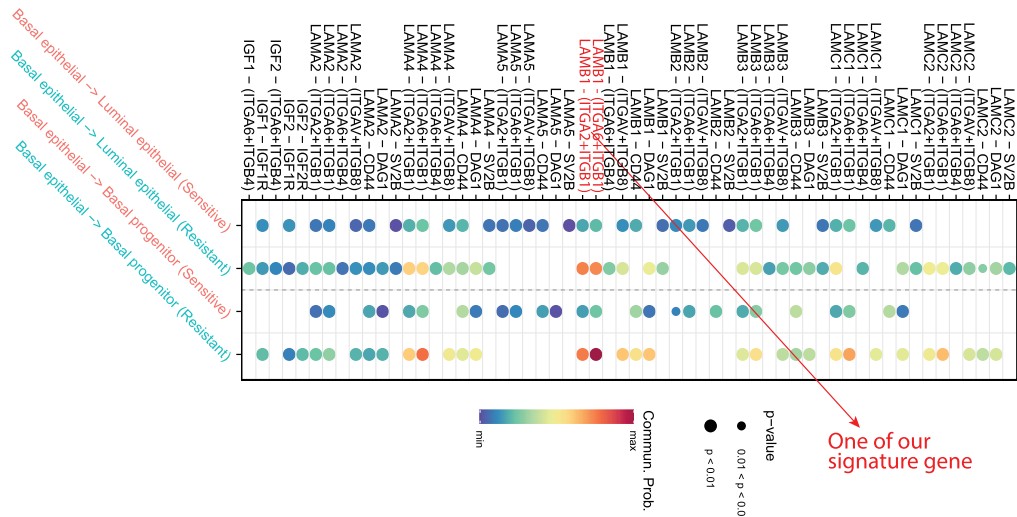

**B.**

Candidate gene matrix

| Gene | Chemoresistance | | BC subtype | | EMT | | Survival (RFS) | |
|---|---|---|---|---|---|---|---|---|
| | scRNA-seq | Bulk RNA-seq | BC Tumors | BC cell lines | EMT-High TNBC (Tumors) | Mammary EMT (HMLE) | Poor survival (TNBC) | Poor survival (LN+ TNBC) |
| ACTG2 | + | + | + | + | + | + | + | + |
| ACTN1 | + | + | + | + | + | + | + | + |
| MYLK | + | + | + | + | + | + | + | + |
| ANXA1 | + | + | + | + | + | + | + | - |
| CNN3 | + | + | + | + | + | + | + | - |
| CAV2 | NA | + | + | + | + | + | + | + |
| MSRB3 | NA | + | + | + | + | + | + | + |
| SFRP1 | + | + | + | + | + | + | - | + |
| TNC | NA | + | + | + | + | + | + | + |
| KRT17 | NA | + | + | + | + | + | + | - |
| ADAMTS1 | NA | + | + | + | + | + | - | + |
| FLNA | NA | + | + | + | + | + | - | + |

**C.**

ACTN1 expression across cancer cell lines (CCLE)

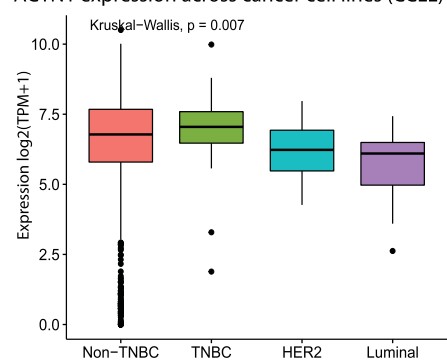

**D.**

Expression of 20 gene signature across integrated dataset

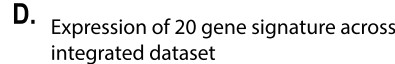
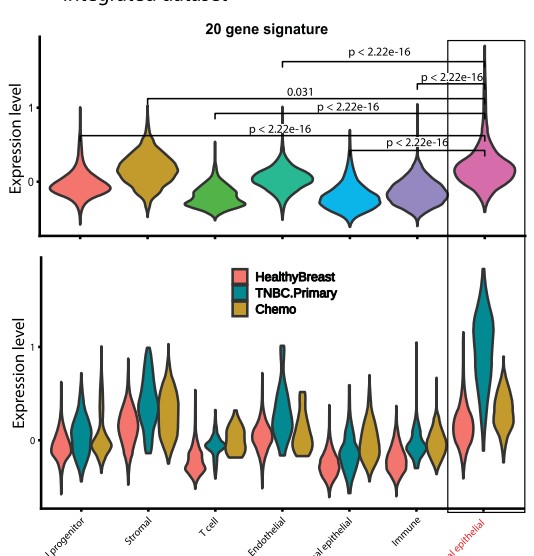

**E.**

Expression of basal and luminal markers across integrated dataset

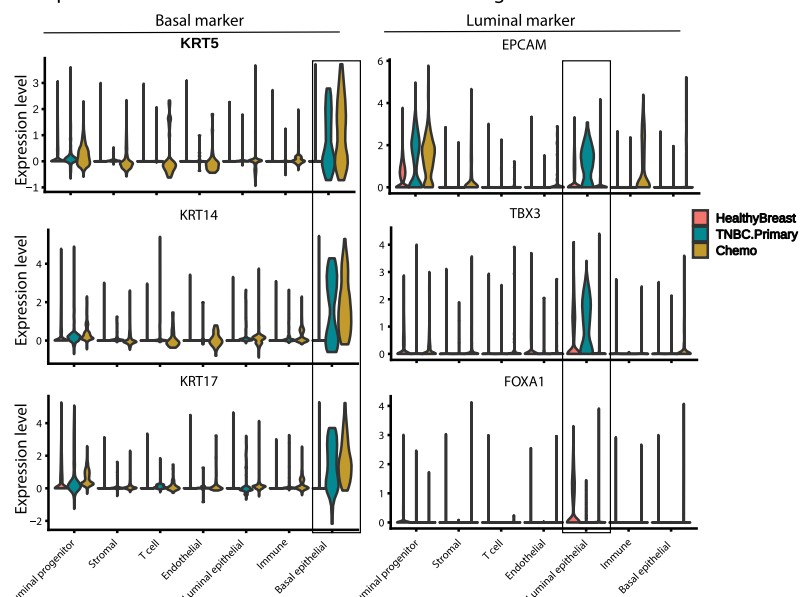

◀ **Figure EV3.** (A) Bar plot showing ligand receptor involved in intercellular signaling of LAMININ signaling pathway. The significance test between signaling pathway signals was computed using two-tailed unpaired Wilcoxon test using the presto package. (B) Table showing candidate gene ranking matrix. The "+" symbol indicates the presence and the "–" sign indicates the absence of a parameter in each of the genes. (C) Expression of *ACTN1* in TNBC, HER2, Luminal, and non-TNBC cell lines. Expression values were obtained from CCLE. The axis shows TNBC, HER2, Luminal, and non-TNBC cancer cell lines and the y axis is their mRNA expression levels. In the box-and-whisker plots, the horizontal lines mark the median, the box limits indicate the 25th and 75th percentiles, and the whiskers extend to 1.5× the interquartile range from the 25th and 75th percentiles. The significance test of expression levels of ACTN1 was performed using Kruskal–Wallis test in ggpubr package. (D) The expression of the 20-gene signature across the cell types shown in upper plot of healthy breast, primary TNBC and chemotherapy-treated TNBC data sets and lower plot shows dataset-wise expression profile. The statistical testing of expression levels of 20 gene was performed using two-tailed unpaired Wilcoxon test in stat_compare_means() of ggpubr package. (E) The expression of basal markers (left violin plots) and luminal epithelial (right violin plot) markers across cell types of the healthy breast, primary TNBC and chemotherapy-treated TNBC cells.

