## [Peer Review File · EMBO Molecular Medicine]

Basal epithelial subpopulations underlie and predict chemotherapy resistance in TNBC

Mohammed Inayatullah, Arun Mahesh, Arran Turnbull, Mike Dixon, Rachael Natrajan, and Vijay Tiwari

Corresponding author(s): Vijay Tiwari (v.tiwari@qub.ac.uk)

Review Timeline:

Submission Date:	11th Nov 23
Editorial Decision:	15th Dec 23
Revision Received:	16th Jan 24
Editorial Decision:	29th Jan 24
Revision Received:	7th Feb 24
Accepted:	14th Feb 24

Editor: Lise Roth

Transaction Report:

(Note: Please note that the manuscript was previously reviewed at another journal and the reports were taken into account in the decision making process at EMBO Molecular Medicine. Since the original reviews are not subject to EMBO's transparent review process policy, the reports and author response cannot be published. With the exception of the correction of typographical or spelling errors that could be a source of ambiguity, letters and reports are not edited. Depending on transfer agreements, referee reports obtained elsewhere may or may not be included in this compilation. Referee reports are anonymous unless the Referee chooses to sign their reports.)

15th Dec 2023

Dear Vijay,

Thank you for the submission of your manuscript to EMBO Molecular Medicine, and please accept my apologies for the delay in getting back to you in this busy time of the year. As you will see below, the referee who was consulted on your revised manuscript, previous reports and rebuttal letter is supportive of publication pending additional clarifications, and I am therefore pleased to inform you that we will be able to accept your manuscript once the following points will be addressed:

1/ Please address the minor comments and questions from the referee and provide a .docx formatted letter INCLUDING the reviewer's report and your detailed point-by-point responses to his/her comments. As part of the EMBO Press transparent editorial process, the point-by-point response is part of the Review Process File (RPF), which will be published alongside your paper

2/ Manuscript text:

- Please provide a .docx formatted version of the manuscript text (including legends for main figures, EV figures and tables). Please make sure that the changes are highlighted to be clearly visible.
- Please remove 'Data not shown' (p13). As per our guidelines on "Unpublished Data", the journal does not permit citation of "Data not shown". All data referred to in the paper should be displayed in the main or Expanded View figures.
- The Methods section should be renamed "Materials and Methods". Please make sure that the information provided matches the author checklist.
- It is mandatory to include a 'Data Availability' section after the Materials and Methods. Before submitting your revision, primary datasets produced in this study need to be deposited in an appropriate public database, and the accession numbers and database listed under 'Data Availability'. Please remember to provide a reviewer password if the datasets are not yet public (see <https://www.embopress.org/page/journal/17574684/authorguide#dataavailability>). Note that the Data Availability Section is restricted to new primary data that are part of this study.
- Acknowledgements: Please make sure that the funding information provided in the manuscript matches the information provided in the submission system.
- Author contributions: CRediT has replaced the traditional author contributions section because it offers a systematic machine-readable author contributions format that allows for more effective research assessment. Please remove the Authors Contributions from the manuscript and use the free text boxes beneath each contributing author's name in our system to add specific details on the author's contribution. More information is available in our guide to authors.
- Please rename "Competing interests" to "Disclosure statement and competing interests": We updated our journal's competing interests policy in January 2022 and request authors to consider both actual and perceived competing interests. Please review the policy <https://www.embopress.org/competing-interests> and update your competing interests if necessary.
- Please reformat the references to have them in alphabetical order, with 10 authors listed before et al.
- Our journal encourages inclusion of *data citations in the reference list* to directly cite datasets that were re-used and obtained from public databases. Data citations in the article text are distinct from normal bibliographical citations and should directly link to the database records from which the data can be accessed. In the main text, data citations are formatted as follows: "Data ref: Smith et al, 2001" or "Data ref: NCBI Sequence Read Archive PRJNA342805, 2017". In the Reference list, data citations must be labeled with "[DATASET]". A data reference must provide the database name, accession number/identifiers and a resolvable link to the landing page from which the data can be accessed at the end of the reference. Further instructions are available at .

3/ Figures:

- Please provide individual production quality figure files as .eps, .tif, .jpg (one file per figure). Supplementary figures should be renamed "Figure EV1" etc... For guidance, download the 'Figure Guide PDF' (<https://www.embopress.org/page/journal/17574684/authorguide#figureformat>).
- Figure 4D is referenced in the text, however the panel is missing in the figure.
- In the figure legends, the heading of the section for the EV figures should be renamed "Expanded View Figure Legends"
- Supplementary Table S1 should be renamed "Table EV1" and a short legend should be added to the top of the page
- Please provide exact p values, not a range, in the figures or in their legends, including for ns - non-significant.
- If images were re-used in different figures (i.e. Figure 1E and Figure 2G), please indicate it in the figure legends.
- Please address the queries from our data editors who are currently checking your figure and figures legends. You should receive their comments shortly.

4/ At EMBO Press we ask authors to provide source data for the main figures. Our source data coordinator will contact you to discuss which figure panels we would need source data for and will also provide you with helpful tips on how to upload and organize the files.

5/ Please provide a complete author checklist, which you can download from our author guidelines (<https://www.embopress.org/page/journal/17574684/authorguide#submissionofrevisions>). Please insert information in the checklist that is also reflected in the manuscript. The completed author checklist will also be part of the RPF.

6/ Please provide 'The paper explained' section: EMBO Molecular Medicine articles are accompanied by a summary of the articles to emphasize the major findings in the paper and their medical implications for the non-specialist reader. Please provide a draft summary of your article highlighting

7/ Every published paper now includes a 'Synopsis' to further enhance discoverability. Synopses are displayed on the journal webpage and are freely accessible to all readers. They include a short stand first (maximum of 300 characters, including space) as well as 2-5 one-sentences bullet points that summarizes the paper. Please write the bullet points to summarize the key NEW findings. They should be designed to be complementary to the abstract - i.e. not repeat the same text. We encourage inclusion of key acronyms and quantitative information (maximum of 30 words / bullet point). Please use the passive voice. Please attach these in a separate file or send them by email, we will incorporate them accordingly.

8/ As part of the EMBO Publications transparent editorial process initiative (see our Editorial at <http://embomolmed.embopress.org/content/2/9/329>), EMBO Molecular Medicine will publish online a Review Process File (RPF) to accompany accepted manuscripts.

This file will be published in conjunction with your paper and will include the anonymous referee reports, your point-by-point response and all pertinent correspondence relating to the manuscript. Let us know whether you agree with the publication of the RPF and as here, if you want to remove or not any figures from it prior to publication.

I look forward to receiving your revised manuscript.

With kind regards,

Lise

***** Reviewer's comments *****

Referee #1 (Comments on Novelty/Model System for Author):

A suitable regression model chosen to select genes for chemotherapy resistance prediction with potential clinical implications. This is the most interesting and original outcome of the paper. Parts of the paper discuss extensively EMT-states in chemoresistance, which is, in my opinion, not so novel.

Referee #1 (Remarks for Author):

In the presented manuscript, the Authors re-analyze published datasets of single-cell RNA sequencing of triple-negative breast cancer (TNBC) patients and find that basal epithelial cells exhibit higher levels of chemoresistance markers. Thus, they derive a signature of 101 shared marker genes (at least across 2 out of 3 used datasets). It is then shown that this signature is enriched also in comparison of Chemoresistant vs Chemosensitive patients' single-cell data both in pre- and post-treatment stage suggesting that this signature might contain predictive potential on chemoresistance. Indeed, the Authors develop predictive model based on regression with elastic net regularization and narrow down the signature to 20 genes which their model uses to predict the chemotherapy response. The regression model is then benchmark against other published predictors and it is shown it has superior performance.

The manuscript contains a large number of results and the Authors integrated many different datasets to prove their points. This is, on one hand, impressive; on the other hand, it is not always easy for the reader to follow the "storyline". I have a couple of remarks and questions, that will hopefully help clarify the flow of the paper.

1. Figure 1B: This figure represents a combination of 3 scRNA-seq datasets from TNBC patients. Which of the cells shown are actually tumor cells? Apart from the clusters which are clearly TME or infiltration (T cells, stroma...) to me the big cluster of luminal epithelial cells in TNBC is quite surprising. My current interpretation is that the Luminal and Basal cells are cancer cells scoring high in the respective markers, but is that correct? CNV inference (perhaps through inferCNV R package) could help show which cells are indeed modified cancer cells, or annotation from the source papers.
2. Figure 1B: How did the Authors correct for batch effect? This is a rather important detail that should be in the Methods.
3. Signature of 101 genes: The Authors decided to overlap marker genes in all three datasets separately and focus on the shared markers. Once the datasets have been integrated (in Fig 1B), wouldn't it be more straightforward to define markers directly from the Basal-like cluster in Fig 1B? Could the Authors comment on that?
4. Figure 1E shows that the population of Basal epithelial cells is uniquely located at the border between the tumor and TME, but the signature is expressed across the whole fibrous region. If the signature is specific, shouldn't its expression correspond to the locations similar to those in the middle panel?
5. Figure 2A and B: Why in this data the Authors do not annotate the cells like in Fig 1A? For example, the significant difference between Chemores. and Chemosens. In Fig2A in the posttreatment setting is clearly driven by a subset of the cells (forming the top "bump" in the violin plots). These might be, again, the basal-epithelial cells perhaps.
6. Figure 3E shows the expression of the EMT markers in the form of Z-score I guess but the Authors refer to the plot as "correlation matrix". This should be clarified. Also, the hierarchical clustering of the samples (vertical dimension) should be shown since by that means the samples are stratified to form the plot Fig 3F.
7. Fig 4B: The selected subtypes are significantly upregulated compared to Low_CIN? The statistics are not shown.
8. Figure 5B: Did the Authors test the capacity of their predictive model also with even more reduced gene set? Naively, looking into the score table, the genes 15 to 20 are less informative than ITGB1 alone. How is actually the Score linked to the LASSO coefficient plot in Fig S3E which depicts the feature selection upon attenuation of the lambda-regulator strength?

I personally like the regression model that the Authors have chosen to build the predictor, I believe it is a suitable choice given the sizes of the training data and number of putative predictors. I also appreciate the benchmarking section.

Minor comments:

1. Fig 1A: consistency in annotation "3,985 cells" and "6862 cells"
2. Suppl Fig 1B and C, typos in the titles of the plots
3. Typo page 20, top word: "...genes were further analyses for..." should be analyzed

RESPONSES TO REVIEWERS

Reviewer 1 (Remarks to the Author):

We thank the reviewer for her/his exciting remarks, **“A suitable regression model chosen to select genes for chemotherapy resistance prediction with potential clinical implications. This is the most interesting and original outcome of the paper.”** The reviewer further comments **“In the presented manuscript, the Authors re-analyze published datasets of single-cell RNA sequencing of triple-negative breast cancer (TNBC) patients and find that basal epithelial cells exhibit higher levels of chemoresistance markers. Thus, they derive a signature of 101 shared marker genes (at least across 2 out of 3 used datasets). It is then shown that this signature is enriched also in comparison of Chemoresistant vs Chemosensitive patients' single-cell data both in pre- and post- treatment stage suggesting that this signature might contain predictive potential on chemoresistance. Indeed, the Authors develop predictive model based on regression with elastic net regularization and narrow down the signature to 20 genes which their model uses to predict the chemotherapy response. The regression model is than benchmark against other published predictors and it is shown is has superior performance.”**

The reviewer had a couple of remarks and questions to help clarify the flow of the paper, which we have implemented as follows:

Comment 1) 1. Figure 1B: This figure represents a combination of 3 scRNA-seq datasets from patients. Which of the cells shown are actually tumor cells? Apart from the clusters which are clearly TME or infiltration (T cells, stroma...) to me the big cluster of luminal epithelial cells in TNBC is quite surprising. My current interpretation is that the Luminal and Basal cells are cancer cells scoring high in the respective markers, but is that correct? CNV inference (perhaps through inferCNV R package) could help show which cells are indeed modified cancer cells, or annotation from the source papers.

Authors' response: We thank the reviewer for this suggestion. We would like to clarify that Figure 1B represents a single scRNA dataset

Figure 1C (source study): Bar plot depicting the distribution of the 1112 cells assigned to specific cell types, by patient. Figure 2A (source study): t-SNE plot of all 1112 classified cells, demonstrating separation of non-epithelial cells by cell type.

derived from Karaayvaz et al, published in Nature communication (Karaayvaz et al, 2018) . The remaining two other scRNA datasets (Chung et al, 2017; Gulati et al, 2020) were processed and analyzed separately (shown in Expanded figure EV1A-C). Our analysis showed that the basal epithelial cells expressed highest levels of genes known to be associated with tumor aggressiveness including metastasis and chemoresistance (**Fig. 1C of submitted and revised manuscript**). We therefore took the genes that overlapped between the basal cells of the three datasets as signature genes for further downstream analysis.

Regarding the observation of a big cluster of luminal epithelial cells, it is in line with previous studies including the source study where most cells were found to be of epithelial identity (Figures 1C and 2A of the source study shown here). The source study has performed further analysis based on clustering, genomic CNVs, and correlation maps that **classified most epithelial cells as malignant cells** (Figure 2a-f of the source study). Importantly however, the authors here did not further classify these epithelial cells into luminal and basal malignant cells.

Figure 2 (source study): Clustering, genomic CNVs, and correlation maps classify most epithelial cells as malignant. a t-SNE plot of all 1112 classified cells, demonstrating separation of non-epithelial cells by cell type. b t-SNE plot of the 244 non-epithelial cells, demonstrating separation by cell type, and no distinguishable patient effect. c t-SNE plot of the 868 epithelial cells, showing mixed separation by patient, and substantial clustering of cells from different patients, suggesting pronounced intra-tumor heterogeneity. d Inferred CNVs from the single-cell gene expression data. Columns represent individual cells, and rows represent a selected set of genes, arranged according to their genomic coordinates (chromosome number indicated at left). A set of 240 normal mammary epithelial cells is shown on the left for comparison, and epithelial cells from all TNBC cases are shown, clustered separately for each patient. Amplifications (red) or deletions (blue) are inferred by computing, for each gene, a 100-gene moving average expression score, centered at the gene of interest. Prominent subclones defined by shared CNVs in tumors 39 and 81 are indicated by brackets on the top ("clonal"). e WES data for four of the six TNBC cases demonstrates high concordance with the CNV calls inferred from the transcriptomes of single cells (d). Genomic coordinates are arranged as in d from top to bottom, and mean copy number for each region ("CNV mean") is indicated on a continuous scale, with red representing gain and blue representing loss. Accordingly, scanning from left (d) to right (e) allows for a comparison of inferred CNVs (d) and actual CNVs (e) for the same regions. f Correlation map among the expression profiles of the normal epithelial cells and the TNBC epithelial cells, depicted in the same order from left to right as d. Normal cells, as well as malignant clonal subpopulations defined by shared CNVs for tumors 39 and 81 (indicated as "clonal" at top), are correlated. The remaining non-clonal epithelial populations in all tumors show relatively poor correlation, supporting their identity as malignant cells.

Prompted by the reviewer's comments, we further investigated this in our data using markers of Luminal and basal epithelial types from the source study (Fig. 3, left heatmap), which confirms the identify of annotated clusters in our analysis (Fig. 3, right violin plots).

Figure 3: Left plots are from source study; Figure 1B, showing expression of cell type markers across single-cells of primary TNBC tumors. The right plot shows the same markers plotted on our clustered cells which aligns with the source study, including markers specific to basal epithelial clusters.

Next, we also plotted other markers of malignancy known in the literature (Hu *et al*, 2023) to stratify malignant cells within the basal and luminal epithelial cell types. Here we have retrieved cancer cell

markers of basal and luminal epithelial types of the breast from cellMarker (Hu *et al.*, 2023) database, which is one of the largest databases of cell type markers originating from different tissues from malignant tumors. Plotting these markers clearly showed that these basal and epithelial clusters exhibit malignant

Known malignant cell markers of Luminal and Basal epithelial type (CellMarker V2.0)

Basal epithelial malignant markers

Luminal epithelial malignant markers

Figure 4: The violin plots show expression of malignant cell markers of Luminal and basal breast cancer type. The cancer cell marker of basal and luminal epithelial type was retrieved from CellMarker database and plotted on our primary TNBC dataset.

characteristics (**Figure 4**). Altogether, these observations confirmed our cell type annotations and that basal and luminal epithelial clusters exhibit malignant characteristics. We have added these results in the revised manuscript (**New figure: Figure EV1D**).

Furthermore, as suggested by the reviewer, we also conducted inferCNV analysis to distinguish malignant cells within the luminal and basal cell types of the epithelial population. Here, a set of 240

normal mammary epithelial cells, as referenced in a previous study (Gao *et al*, 2017), served as a benchmark to differentiate between malignant and non-malignant epithelial cells. Subsequently, a total of 860 epithelial cells were analyzed in our TNBC dataset, comprising 602 luminal epithelial cells, 188 luminal progenitor cells, and 70 basal epithelial cells, and copy number changes were computed by comparing with normal mammary epithelial cells using inferCNV package.

Our analysis unequivocally reveals that the majority of epithelial cells, including basal epithelial cells, exhibit an altered copy number variation (CNV) profile, as depicted in the bottom heatmap (**Figure 5, bottom heatmap**). This divergence is particularly notable when compared to the reference normal epithelial cells illustrated in the upper heatmap (**Figure 5, top heatmap**). This observation strongly suggests a malignant nature for these cells, aligning with findings from a source study that predominantly classified epithelial cells within the triple-negative breast cancer (TNBC) dataset as malignant cells (Karaayvaz *et al.*, 2018) (see Figure 2a-f in the source study). These data have been included in the revised manuscript (**New figure: Figure 1D**).

Figure 5: The infercnv analysis classified majority of TNBC cells as malignant cells. The upper heatmap plot shows copy number alteration profile in healthy mammary epithelial cells. The lower heatmap plot showing CNV profile of TNBC epithelial cells. We have used total 240 healthy mammary epithelial cells to compute copy number alteration in TNBC epithelial, including basal epithelial cells.

The upper heatmap was generated by subtracting the expression profile of normal epithelial cells from the expression data of TNBC epithelial cells, highlighting differences. Regions of chromosomal amplification manifest as blocks of red, while chromosomal deletions manifest as blue blocks, providing a visual representation of the copy number changes.

In addition to the above CNV heatmap, we further confirmed malignant nature of these cells by scoring each cell based on the extent of CNV signal identified through inferCNV. This scoring was derived from the number of genes exhibiting copy number alterations (CNA) in each cell, as obtained from infercnv_obj@expr.data in the inferCNV output. Subsequently, putative malignant cells were discerned based on their inferCNV scores. Lower scores signified a diminished CNV signal, while higher scores indicated a heightened CNV signal within the cells. Plotting these scores across all cells revealed a binomial distribution centered around an infercnv score of 0.2 (**Figure 6A**). Notably, cells scoring less than 0.2 predominantly comprised normal mammary epithelial cells, whereas cells scoring above 0.2 predominantly identified as triple-negative breast cancer (TNBC) epithelial cells (**Figure 6A and B**). This distribution was further dissected across each cell type, underscoring that the majority of TNBC cells, including basal epithelial cells, exhibited a heightened CNV signal

Malignant cell classification based on Infercnv CNV outcome (InferCNV score >0.2 are malignant; InferCNV score <0.2 are Normal)

Figure 6: Copy number score computed from inferCNV shows clear separation of TNBC epithelial vs normal epithelial cells, indicating TNBC cells as malignant cells. A The histogram plot shows binomial distribution of infercnv score of normal epithelial vs TNBC epithelial cells. The infercnv scores less than 0.2 defined normal epithelial cells and score greater than 0.2 defined TNBC epithelial cell types. **B** boxplot depicting distribution of infercnv scores of cells having significant difference between the normal epithelial vs TNBC epithelial cells. **C** and **D** shows similar infercnv score profile across each cell types. The red dotted lines shows infercnv scores threshold separating normal epithelial from TNBC epithelial cells.

compared to normal mammary epithelial cells (**Figure 6C and D**). These data have been included in the revised manuscript (**New figure: Figure 1E**).

In summary, these observations clearly demonstrate that most of the TNBC epithelial cells, including basal epithelial cells, are malignant. These findings also align with the source study (Karaayvaz *et al.*, 2018), where the majority of epithelial cells were classified as malignant cells. We thank the reviewer for motivating us to perform these analyses as it has added new, relevant supporting data to our manuscript.

Comment 2) Figure 1B: How did the Authors correct for batch effect? This is a rather important detail that should be in the Methods.

Author's response: We agree with the reviewer that batch correction is a very critical step in such a workflow, and hence we paid serious attention to this prior to the analysis. Here we corrected the batch effect using the established canonical correlation analysis (CCA) method in Seurat and mentioned this in our Methods section as follows: "Batch effect across multiple samples were regress out and the integration of scRNA-seq datasets were done using canonical correlation analysis (CCA) method in Seurat. As a reflection of a successful batch correction, it is clearly seen that cells are clustered based on the cell type and not patient samples (**Figure 7**).

Here, within primary TNBC dataset (used in figure 1 of submitted manuscript), we could not see any batch effect, as we see each cluster is contributed by multiple samples and annotated as distinct cell types (**Figure 7**).

Karaayvaz et al, 2019 dataset used in **Figure 1**

Figure 7: The umap plot is of primary TNBC samples shows, there are no batch effects exists, as cells clustered based on distinct celltypes and not by individual patient samples.

However, for scRNA-seq dataset (used in figure 2 of submitted manuscript) from Kim et al, showed a strong batch effect in both resistant and sensitive datasets (**Figure 8, Upper Umap plot**), which we removed using CCA method in Seurat. It is evident that after batch effect correction, the cells are clustered based on cell-types in both resistant and sensitive datasets (**Figure 8, Lower Umap plot**). We have added these details in the expanded figure EV2A of the manuscript as a new figure (**New figure:Figure EV2A**).

Kim et al, 2018 dataset used in **Figure 2**

Figure 8: The upper UMAP plot shows existence of possible batch effect in resistant and sensitive datasets. The batch effects were regressed out using CCA and samples were integrated in Seurat. The bottom UMAP plots clearly shows cells are clustered based on the cell type and hence shows removal of possible batch effects from the datasets.

Comment 3) Signature of 101 genes: The Authors decided to overlap marker genes in all three datasets separately and focus on the shared markers. Once the datasets have been integrated (in Fig 1B), wouldn't it be more straightforward to define markers directly from the Basal-like cluster in Fig 1B? Could the Authors comment on that?

Author's response: We thank the reviewer for this comment. We apologize for not making it clear that the data shown in Fig. 1B is from a single study and not integration of all 3 independent datasets. In fact, we processed and analyzed all 3 datasets individually and overlapped the gene sets of their respective basal epithelial cluster to identify robust gene signatures of tumor aggressiveness.

Nevertheless, as suggested by reviewer, we have checked whether our gene signature remains intact when we perform these analysis on the integrated single-cell datasets. The clustering of cells showed batch effects as cells clustered based on the datasets (**Figure 9A, left umap**) which was removed using the CCA method for integration as explained in the earlier comment (**Figure 9B, left umap**) as cells from independent datasets contributed to each cell cluster and annotated as

Figure 9: Integration of 3 scRNA-seq data analysis shows basal epithelial signature intact, like independent analysis approach. **A)** UMAP plot shows clustering of cells from 3 independent datasets and possible batch effect, as samples clustered based on the dataset. **B)** The UMAP plot shows good integration and batch effect removal after applying canonical correlation analysis (CCA) method on the dataset. The cells are clustered based on the cell types and shared from independent datasets, hence indicate batch effect removal from the datasets. **C)** shows expression of aggressive gene signatures from earlier studies also used in our figure 1 C of the revised manuscript. the mean expression of 49 metastasis and 143 chemoresistance signature genes were plotted on the integrated datasets and it shows higher enrichment within basal cluster compared to other cell types. **D)** The UMAP shows the intactness of our signature genes (101 genes from Figure 1G) as well as our predictive gene signature of pCR and RD within basal epithelial cluster. We could see, in line with our independent approach, these signatures are enriched only in basal epithelial cells in the integrated datasets.

distinct cell types (**Figure 9B, right umap**). Interestingly, in line with our previous findings, we again found the cells of basal epithelial type highly enriched with both metastasis (**Figure 9C, left umap**) and chemoresistance (**Figure 9C, right umap**) gene signatures as compared to other cell types in this well-integrated data. Furthermore, our gene signature of 101 genes largely remained intact in basal epithelial cells in the integrated dataset and not enriched in other cell types (**Figure 9D, left umap**). Along these lines, our refined and predictive 20 gene signature was still exclusively enriched in basal epithelial cells (**Figure 9D, right umap**). This new investigation using integrated datasets further confirmed the authenticity of our previous approach and demonstrates the robustness of our gene signature. We thank the reviewer for this suggestion as this has provided additional validations to our study.

Comment 4) Figure 1E shows that the population of Basal epithelial cells is uniquely located at the border between the tumor and TME, but the signature is expressed across the whole fibrous region.

If the signature is specific, shouldn't its expression correspond to locations similar to those in the middle panel?

Author's response: We thank reviewer for raising this query. This led us to revisit the spatial data and corresponding analysis and found an issue with the normalization method. Earlier we had used SCTransform() method, which potentially diminished minor differences in intensity across cells within the tissue. Along these lines, a recent study on spatial transcriptome analysis benchmarking have demonstrated that sctransform normalization works poorer as compared to log normalization for spatial transcriptome deconvolution (Li et al, 2023). To overcome this, we have now normalized these data with LogNormalize() method and subsequently plotted expression of our

Figure 10: The spatial analysis of TNBC tissue section shows spatial arrangement of basal epithelial cells in close vicinity to stromal compartment. **A** The left plot shows tissue section of aggressive TNBC tumor, the right right plot shows cell type deconvolution of TNBC cell types within histological section. **B** the left plot shows mean expression of our signature within the tissue section and right plot boxplot is showing expression levels of our gene signature across different cell types of the same spatial dataset.

signature genes and which showed a much clearer enrichment of our signature genes within basal epithelial cells compared to other cell types on the tissue section (**Figure 10A and B, left plot**). These observations were further validated by quantifications, where we found a significant elevation in expression levels of signature genes in basal epithelial cells compared to other cell types within the spatial dataset (**Figure 10B, right violin plot**).

Comment 5) Figure 2A and B: Why in this data the Authors do not annotate the cells like in Fig 1A? For example, the significant difference between Chemores. And Chemosens. In Fig 2A in the posttreatment setting is driven by a subset of the cells (forming the top "bump" in the violin plots). These might be, again, the basal-epithelial cells perhaps.

Author's response: We thank the reviewer for this suggestion. Following this, we have performed cell type annotations of chemoresistant and chemosensitive cells in the suggested plots (**Figure 11A and B**), which showed that our signature is more highly enriched in the basal epithelial populations (**Figure 11C, left**). Therefore, the reviewer's assumption is right, as the post-treated cells in chemoresistant group are enriched with our gene signature expression within basal cells that form the top "bump" in the violin plots in the previous version (Figure 2C and D of revised manuscript). We have added these findings to the revised manuscript that has further increased the comprehensibility and impact of our study (**New figure: Figure 2A-D**).

Figure 11: Cell type annotation of chemoresistant and chemosensitive tumors shows enrichment of our signature genes in post treated chemoresistant tumors. A-B. Left and right UMAP plot shows cell type annotation of chemosensitive and chemoresistant cells. C. left UMAP plot shows expression of our signature genes in chem resistant dataset. Right violin plot shows expression of our signature across different cell types of chemoresistant tumor dataset.

Comment 6) Figure 3E shows the expression of the EMT markers in the form of Z-score I guess but the authors refer to the plot as "correlation matrix". This should be clarified. Also, the hierarchical clustering of the samples (vertical dimension) should be shown since by that means the samples are stratified to form the plot Fig 3F.

Authors' response: We apologize for the typo in the legend for Fig 3E. Indeed the expression of EMT markers is shown as a Z-Score and we have now corrected it accordingly in the legend. Furthermore, as suggested by the Reviewer, we have now also performed hierarchical clustering of the samples (vertical dimension) in Fig. 3E that showed a clear clustering of epithelial and mesenchymal markers (**Figure 12**). We thank the reviewer for these suggestions. We have updated this figure in the revised manuscript.

Figure 12: Heatmap shows expression profiling of hallmark epithelial and mesenchymal genes across TCGA TNBC tumors. We have used expression of 4 epithelial and 6 mesenchymal markers to classify TNBC tumors into EMT high (Mesenchymal), Hybrid and EMT-Low (Epithelial) like tumors.

Comment 7) Fig 4B: The selected subtypes are significantly upregulated compared to Low_CIN? The statistics in not shown.

Author's response: In the figure 4B, we investigated the expression of our signature genes among six previously defined CNA subtypes of TNBC (Jiang *et al*, 2019). These CNA subtype represents CNA subtype 1, frequent 9p23 amplification (*Chr9p23 amp*); CNA subtype 2, frequent 12p13 amplification (*Chr12p13 amp*); CNA subtype 3, frequent Chr13q34 amplifications (*Chr13q34 amp*); CNA subtype 4, frequent Chr20q13 amplification (*Chr20q13 amp*); CNA subtype 5, frequent Chr8p21 loss (*Chr8p21 del*); and CNA subtype 6, somatic CNA lacking a CN cluster but with low chromosomal instability (CIN) (*low CIN*). Indeed, our signature genes were not significantly upregulated within any of these groups, but showed a trend of elevation in the tumors of frequently amplified group subtypes compared to Low CIN groups (**Figure 13 left boxplot**). Consequently, we did not claim in the manuscript that this difference was 'significant'.

Figure 13: The left boxplot shows expression of our signature within high copy number vs low CIN groups. The right plot shows expression of our signature genes in mutation type categories in TNBC.

However, when investigated in the mutation type categories (**shown in Figure 4C of the submitted manuscript**), our signature genes did show a significant elevation in expression in tumors with Homologous Recombination Deficiency (HRD) compared to other mutation types in TNBC (**Figure 13 right boxplot**).

Comment 8) Figure 5B: Did the Authors test the capacity of their predictive model also with an even more reduced gene set? Naively, looking into the score table, the genes 15 to 20 are less informative than ITGB1 alone. How is actually the Score linked to the LASSO coefficient plot in Fig S3E which depicts the feature selection upon attenuation of the lambda-regulator strength?

Authors' response: Indeed, we had tested the capacity of our predictive model with further reduced

gene sets as suggested by the reviewer. We have removed genes that ranked 15 to 20 (**Figure 5B of the submitted version and Figure 5C of revised version**) and evaluated their effect on model performance. Expectedly, the removal of these low-ranked genes could decrease the performance of upto ~3%, (AUC=87.7%) indicating minimal effect on the model performance (**Figure 14**).. Though with the top 14 genes we could

Figure 14: The right plot shows lasso coefficient values based ranking of features for predictive model building. The left ROC plot shows performance evaluation of our predictive classifier upon removal of genes ranked 15 to 20.

achieve an accuracy of upto 87% AUC (which is considered to be a good model performance), we wanted to further enhance the performance with inclusion of further genes, resulting in the final 20-gene signature. Overall, these observations show that our 20-gene signature is a perfect combination of genes for predicting chemotherapy response and any compromise would see a dramatic decrease in this predictive power (**Figure 14**).

In traditional linear regression models might encounter overfitting, wherein the model becomes excessively intricate and closely tailors itself to the training data, leading to poor performance when applied to new data. Lasso regression can help to address this problem by identifying the most important variables and reducing the complexity of the model. For the LASSO coefficient plot in Fig S3E of submitted manuscript, each curve corresponds to a variable (each curve is 1 gene of our 101 signature). It shows the path of its coefficient against the ℓ_1 -norm of the whole coefficient vector as

λ varies. The axis above indicates the number of nonzero coefficients at the current λ , which is the effective degrees of freedom (df) for the lasso. In the plot from left to right, we observe that at first (Figure 15), the lasso models contain many predictors with high magnitudes of coefficient estimates.

With increasing lambda, the coefficient estimates approximate towards zero. In simple terms, the curves that have high magnitudes of coefficient estimates are stronger predictors of the models, compared to the ones which are close to zero coefficient values. Next ranking of lasso coefficient values of these high-magnitude features was performed using varImp() function in caret R package,

Top ranked features

Gene	Score
ITGB1	3.014272
RBFOX2	1.029056
DST	0.805166
RCAN1	0.73652
C9orf3	0.727666
ACTA2	0.702723
S100B	0.601085
LYGE	0.59008
CTNNA1	0.476695
PRNP	0.462167
TIMP3	0.425849
CD63	0.418899
IFI16	0.411074
NFIB	0.393945
ACTN1	0.326654
SFRP1	0.258573
STOM	0.256755
COL6A1	0.24101
DSC3	0.17199
AMIGO2	0.095426

Feature importance scoring
 Standardized scaling
 varImp() function in caret R package

Figure 15: The coefficients from the Lasso fit represent the contributions of the 20 genes expression in the model. The right plot shows lasso regression coefficient values in which each curve corresponds to a variable. It shows the path of its coefficient against the Log Lambda of the whole coefficient vector as λ varies. The axis above indicates the number of nonzero coefficients at the current λ , which is the effective degrees of freedom (df) for the lasso. Each variable with higher magnitude of lasso coefficient were subjected for scaling using caret R package assessing and ranking feature importance.

which creates the standardized scale of final coefficients of the fit and signifies feature importance.

We have provided the code below for your reference on how we computed the feature score.

```

varImp <- function(object, lambda = NULL, ...) {
  beta <- predict(object, s = lambda, type = "coef")
  if(is.list(beta)) {
    out <- do.call("cbind", lapply(beta, function(x) x[,1]))
    out <- as.data.frame(out)
  } else out <- data.frame(Overall = beta[,1])
  out <- abs(out[rownames(out) != "(Intercept)",,drop = FALSE])
  out <- out/max(out)
  out[order(out$Overall, decreasing = TRUE),,drop=FALSE]
}
varImp(cv.lassoModel, lambda = cv.lassoModel$lambda.min, scale=T)

```

References:

- Chung W, Eum HH, Lee HO, Lee KM, Lee HB, Kim KT, Ryu HS, Kim S, Lee JE, Park YH *et al* (2017) Single-cell RNA-seq enables comprehensive tumour and immune cell profiling in primary breast cancer. *Nat Commun* 8: 15081
- Gao R, Kim C, Sei E, Foukakis T, Crosetto N, Chan LK, Srinivasan M, Zhang H, Meric-Bernstam F, Navin N (2017) Nanogrid single-nucleus RNA sequencing reveals phenotypic diversity in breast cancer. *Nat Commun* 8: 228
- Gulati GS, Sikandar SS, Wesche DJ, Manjunath A, Bharadwaj A, Berger MJ, Ilagan F, Kuo AH, Hsieh RW, Cai S *et al* (2020) Single-cell transcriptional diversity is a hallmark of developmental potential. *Science* 367: 405-411
- Hu C, Li T, Xu Y, Zhang X, Li F, Bai J, Chen J, Jiang W, Yang K, Ou Q *et al* (2023) CellMarker 2.0: an updated database of manually curated cell markers in human/mouse and web tools based on scRNA-seq data. *Nucleic Acids Res* 51: D870-D876
- Jiang YZ, Ma D, Suo C, Shi J, Xue M, Hu X, Xiao Y, Yu KD, Liu YR, Yu Y *et al* (2019) Genomic and Transcriptomic Landscape of Triple-Negative Breast Cancers: Subtypes and Treatment Strategies. *Cancer Cell* 35: 428-440 e425
- Karaayvaz M, Cristea S, Gillespie SM, Patel AP, Mylvaganam R, Luo CC, Specht MC, Bernstein BE, Michor F, Ellisen LW (2018) Unravelling subclonal heterogeneity and aggressive disease states in TNBC through single-cell RNA-seq. *Nat Commun* 9: 3588
- Li H, Zhou J, Li Z, Chen S, Liao X, Zhang B, Zhang R, Wang Y, Sun S, Gao X (2023) A comprehensive benchmarking with practical guidelines for cellular deconvolution of spatial transcriptomics. *Nat Commun* 14: 1548

29th Jan 2024

Dear Vijay,

Thank you for the submission of your revised manuscript to EMBO Molecular Medicine. We have now received the feedback from the referee who was consulted on your manuscript. As you will see below, he/she is supportive of publication, and I will therefore be able to accept your manuscript once the following editorial points will be addressed:

1/ Manuscript text:

- Please accept previous changes, and only keep in track changes mode any new modification.
- Materials and Methods:
 - o Cell culture: please indicate the origin of the cells, and whether they were authenticated and tested for mycoplasma contamination.
 - o Primers sequences should be in the main manuscript.
 - o Please add a statistics section with mention of blinding, sample size, randomization, etc. (please refer to the authors checklist).
- Data Availability section: please add an URL link to the dataset.
- Please correct the order of the following sections to: Disclosure and competing interests statement, References, Figure legends, Tables and their legends, Expanded View Figure legends.
- Please remove "Supplementary information".
- Please remove the legend for Table EV1 from the manuscript file and add it to the Excel file.
- Figure legends: For EV figures, the section heading should be "Expanded View Figure Legends", and the figures should be named "Figure EV1", etc.
- Data citation: please incorporate the Data citations references to the rest of the references (in alphabetical order).

2/ Figures:

- Please provide exact p values for Figure EV3 panel D.
- Figure 8I contains error bars based on n=2. Please use scatter blots showing the individual datapoints in these cases. The use of statistical tests needs to be justified.

4/ Thank you for providing Source Data. Please upload them as one file per figure.

5/ Checklist:

Please complete/correct the following sections:

- Primers sequences
- Cell authentication and mycoplasma
- Experimental study design and statistics.

6/ The Paper Explained: I introduced minor modifications in your text, please let me know if you agree or amend as you see fit:

Problem

Triple-Negative Breast Cancer (TNBC) is the most aggressive type of breast cancer and is hard to treat. It spreads fast, doesn't respond well to chemotherapy, and often leads to poor outcomes in patients. Despite advances in the field, the molecular basis of these aggressive behaviors remains poorly understood.

Results

In this study, we used advanced techniques that allow a closer look at the TNBC tumor cells and their genes at the single-cell and spatial resolution. This analysis identified specific groups of cells in the tumor that exhibit resistance to chemotherapy. Furthermore, these cells express certain genes that are highly active and predictive of future response to chemotherapy. Interestingly, high levels of ITGB1 improves cell communication, and ACTN1 expression gives cells a survival advantage, fostering resistance to chemotherapy. Furthermore, we identified existing drugs that may be repurposed against chemoresistant tumors.

Impact

Our findings provide an explanation on why certain TNBC tumors are resistant to chemotherapy and proposes a biomarker for predicting patient's response to chemotherapy. This work opens avenues for precision medicine, providing stratification biomarker and alternative therapies for better managing TNBC patients resistant to traditional chemotherapy.

7/ I introduced minor modifications in your synopsis text, please let me know if you agree or amend as you see fit:

Chemotherapy resistance is a key challenge in Triple-Negative Breast Cancer (TNBC). Combining single-cell, spatial and bulk transcriptome analysis with machine learning, we uncovered mechanisms of TNBC chemoresistance that provide biomarkers for chemotherapy response and novel avenues for therapy.

- Basal-epithelial subpopulations underlie chemoresistance in TNBC.
- Chemoresistance-associated basal-epithelial cells reside in close vicinity to stromal compartments within TNBC tumors and engage in enhanced intercellular communication.
- These subpopulations are defined by distinct signature genes that provide the best-in-class predictive biomarker of chemotherapy response.
- Drug repurposing analysis identified existing FDA-approved drugs that may benefit chemoresistant patients.

8/ Please let us know whether you agree with the publication of the Review Process File, and as here, or if you want to remove any figure. As mentioned previously, the RPF would only include reviewer comments and information related to the peer review at EMBO Press. Any information prior to transfer would not be part of this file.

I look forward to receiving your revised manuscript.

With kind regards,

Lise

To submit your manuscript, please follow this link:
<https://embomolmed.msubmit.net/cgi-bin/main.plex>

***** Reviewer's comments *****

Referee #1 (Comments on Novelty/Model System for Author):

Relevant question to be asked (from medical perspective), well chosen regression model, extensive benchmarking. To prove the the "real" medical impact of the proposed predictor, future work is still needed.

Referee #1 (Remarks for Author):

I thank the Authors for their extensive response and clarifications.

The authors addressed the remaining editorial issues.

14th Feb 2024

Dear Vijay,

Thank you for sending the revised files. I am pleased to inform you that your manuscript is accepted for publication and is now being sent to our publisher to be included in the next available issue of EMBO Molecular Medicine.

We note that there is an additional panel in the new Figure EV3, please carefully check the file and send us the corrected figure as soon as possible.

With kind regards,

Lise
